



**ModIs Dust AeroSol (MIDAS): A global fine resolution dust optical depth dataset**
Antonis Gkikas[1], Emmanouil Proestakis[1], Vassilis Amiridis[1], Stelios Kazadzis[2,3], Enza Di Tomaso[4],
Alexandra Tsekeri[1], Eleni Marinou[5], Nikos Hatzianastassiou[6] and Carlos Pérez García-Pando[4,7]
[1]Institute for Astronomy, Astrophysics, Space Applications and Remote Sensing, National Observatory of Athens,
Athens, 15236, Greece
[2]Physikalisch-Meteorologisches Observatorium Davos, World Radiation Center, Switzerland
[3]Institute of Environmental Research and Sustainable Development, National Observatory of Athens, Greece
[4]Earth Sciences Department, Barcelona Supercomputing Center, Barcelona, Spain
[5]Deutsches Zentrum für Luft- und Raumfahrt (DLR), Institut für Physik der Atmosphäre, Oberpfaffenhofen, Germany
[6]Laboratory of Meteorology, Department of Physics, University of Ioannina, Ioannina, Greece
[7]ICREA, Passeig Lluís Companys 23, 08010 Barcelona, Spain
Corresponding author: Antonis Gkikas (agkikas@noa.gr)
**Abstract**
Monitoring and describing the spatiotemporal variability of dust aerosols is crucial to understand
their multiple effects, related feedbacks and impacts within the Earth system. This study describes
the development of the MIDAS (ModIs Dust AeroSol) dataset. MIDAS provides columnar daily dust
optical depth (DOD at 550 nm) at global scale and fine spatial resolution (0.1° x 0.1°) over a decade
(2007-2016). This new dataset combines quality filtered satellite aerosol optical depth (AOD)
retrievals from MODIS-Aqua at swath level (Collection 6, Level 2), along with DOD-to-AOD ratios
provided by MERRA-2 reanalysis to derive DOD on the MODIS native grid. The uncertainties of
MODIS AOD and MERRA-2 dust fraction with respect to AERONET and CALIOP, respectively,
are taken into account for the estimation of the total DOD uncertainty (including measurement and
sampling uncertainties). MERRA-2 dust fractions are in very good agreement with CALIOP column-
integrated dust fractions across the "dust belt", in the Tropical Atlantic Ocean and the Arabian Sea;
the agreement degrades in North America and the Southern Hemisphere where dust sources are
smaller. MIDAS, MERRA-2 and CALIOP DODs strongly agree when it comes to annual and
seasonal spatial patterns; however, deviations of dust loads' intensity are evident and regionally
dependent. Overall, MIDAS is well correlated with ground-truth AERONET-derived DODs
(R=0.882), only showing a small negative bias (-0.009 or -5.307%). Among the major dust areas of
the planet, the highest R values (up to 0.977) are found at sites of N. Africa, Middle East and Asia.



MIDAS expands, complements and upgrades existing observational capabilities of dust aerosols and it is suitable for dust climatological studies, model evaluation and data assimilation.

## 1. Introduction

Among tropospheric and stratospheric aerosol species, dust aerosol is the most abundant component in terms of mass, contributing more than half to the global aerosol amount (Textor et al., 2006; Zender et al., 2011). Preferential sources of dust aerosol are located in areas where precipitation is low, thus favoring aridity, whereas a significant contributing factor is the accumulation of alluvial sediments. Such regions comprise deserts, dry lake beds and ephemeral channels (e.g., Middleton and Goudie, 2001; Prospero et al., 2002; Ginoux et al., 2012). Previous studies (Prospero et al., 2002; Ginoux et al., 2012), have shown that the major portion of the global dust burden originates from the Sahara Desert, which hosts the most intense dust source of the planet, the Bodélé Depression located in the northern Lake Chad Basin. In North Africa, large amounts of mineral particles are also emitted in the Western Sahara while other noticeable sources of smaller spatial extension are located in the eastern Libyan Desert, in the Nubian Desert (Egypt) and Sudan (Engelstaedter et al., 2006).

One of the major dust sources of the planet, following N. Africa, is the Middle East with several active regions (Pease et al., 1998; Hamidi et al., 2013; Yu et al., 2013) in which wind-blown dust is emitted from alluvial plains (Tigris-Euphrates River) and sandy deserts (Rub al Khali Desert). Important dust sources are also recorded in the Asian continent, particularly in the Taklamakan Desert (Ge et al., 2014), in the Gobbi Desert (Chen et al., 2017), in its central parts (Karakum Desert; Li and Sokolik, 2018), in the Sistan Basin (Alizadeh Choobari et al., 2013) and in desert areas (e.g. Thar Desert) situated in the Indus valley plains of Pakistan (Hussain et al., 2005). In North America, mineral particles emitted from the Mojave and Sonoran deserts (Hand et al., 2017) have mainly natural origin while in the Chihuahuan Desert as well as in the Southern Great Plains the anthropogenic interference on soil can favor emission of dust particles and subsequently their entrainment in the atmosphere (Hand et al., 2016). Overall, the major portion of the global dust budget arises from the deserts of the N. Hemisphere (Ginoux et al., 2012) while mineral aerosols are also emitted in Australia (Ekström et al., 2004), South Africa (Bryant et al., 2007; Vickery et al., 2013) and South America (Gassó and Torres, 2019), but to a lesser extent. At global scale, most of the entrained dust loads in the atmosphere originate from tropical and sub-tropical arid regions; however, about 5% of the global dust budget consists of particles emitted from high-latitude sources (Bullard et al., 2016).





Dust plays a key role in several aspects of the Earth system such as climate (e.g. Lambert et al.,
2013; Nabat et al., 2015) and weather (Pérez et al., 2006; Gkikas et al., 2018; Gkikas et al., 2019),
attributed to the perturbation of the Earth-Atmosphere system radiation budget (Sokolik and Toon,
1996; Haywood and Bucher, 2000) by mineral particles, the productivity of oceanic waters (Jickells
et al., 2005) and terrestrial ecosystems (Okin et al., 2004), and humans' health (Kanatani et al., 2010;
Kanakidou et al., 2011; Pérez García-Pando et al., 2014; Du et al., 2016). Dust is characterized by a
pronounced temporal and spatial variability due to the heterogeneity of the emission, transport and
deposition processes governing the dust life cycle (Schepanski, 2018). A variety of atmospheric
circulation mechanisms, spanning from local to planetary scales, are responsible for the uplifting of
erodible particles from bare soils (Koch and Renno, 2005; Knippertz et al., 2007; Klose and Shao,
2012; Fiedler et al., 2013) and their subsequent transport (Husar et al., 2001; Prospero and Mayol-
Bracero, 2013; Yu et al., 2015; Flaounas et al., 2015; Gkikas et al., 2015), accumulation and removal
(Zender et al., 2003; Ginoux et al., 2004) from the atmosphere.
Given the scientific importance of dust in the Earth system as well as the numerous socioeconomic
impacts (Stefanski and Sivakumar, 2009; Weinzierl et al., 2012; Kosmopoulos et al., 2018), there is
a need to monitor and forecast dust loads at different spatiotemporal scales. Contemporary satellite
observations, available over long-term periods, have been proven a powerful tool in such efforts as
they provide wide spatial coverage, relatively high sampling frequency and considerably high
accuracy. Spaceborne retrievals have been widely applied in aerosol research for the description of
dust loads' features and their evolution (e.g., Kaufman et al., 2005; Liu et al., 2008; Peyridieu et al.,
2013; Rashki et al., 2015; Gkikas et al., 2013; 2016; Marinou et al., 2017; Proestakis et al., 2018).
Even more accurate aerosol observations, but locally restricted, are derived by ground-based
platforms consisting of sunphotometers, lidars and in-situ instruments. Based on these measurements,
columnar optical and microphysical properties of mineral particles have been analyzed extensively
(Giles et al., 2012), altitude-resolved information of optical properties has provided insight about the
dust vertical distribution (Mamouri and Ansmann, 2014), and a comprehensive description of dust
optical, microphysical and chemical properties has been achieved from surface and aircraft in-situ
instruments (Rodríguez et al., 2012; Liu et al., 2018). Finally, through the deployment of atmospheric-
dust models (e.g., Pérez et al., 2011; Haustein et al., 2012), global (e.g., Ginoux et al., 2004) and
regional (e.g., Basart et al., 2012) displays of dust burden have been realized.
Traditionally, observations have been utilized to evaluate and eventually constrain model
performance. Observations are increasingly used in data assimilation (DA) schemes for aerosol
forecast initialization (Di Tomasso et al., 2017) and development of reanalysis datasets (Benedetti et
al., 2009; Lynch et al., 2016; Gelaro et al., 2017). The most exploited reanalysis datasets in dust-



related studies, are the MERRAero (Modern Era Retrospective analysis for Research and
Applications Aerosol Reanalysis; Buchard et al., 2015) and its evolution MERRA-2 (Modern-Era
Retrospective analysis for Research and Applications, Version 2; Gelaro et al., 2017) as well as
CAMSRA (Copernicus Atmosphere Monitoring Service Reanalysis; Innes et al., 2019) and its
predecessor MACC (Monitoring Atmospheric Composition and Climate; Inness et al., 2013). Current
reanalysis datasets provide information about dust aerosols at high temporal resolution and decadal
time scales. However, even though aerosol optical depth (AOD) observations are assimilated, the
performance of the simulated outputs is partly model-driven and their resolution is relatively coarse.
The overarching goal of the present study is to describe the development of the MIDAS (ModIs
Dust AeroSol) dataset, which provides dust optical depth (DOD) over a decade (2007-2016). The
powerful element of this product is its availability at fine spatial resolution (0.1° x 0.1°) as well as
the provision of full global coverage, i.e. both over land and ocean. Ginoux et al. (2012) analyzed
DOD at the same spatial resolution and for a long-term period but they restricted only above
continental surfaces since their scientific focus was the identification of natural and anthropogenic
dust sources. Voss and Evan (2020) combined satellite (MODIS, AVHRR) aerosol retrievals and
MERRA-2 winds, and analyzed DOD over long-term periods at coarse spatial resolution (1° x 1°).
Vertical dust (and other aerosol) backscatter and extinction profiles along with the respective column
integrated AODs at 1° x 1° spatial resolution are distributed through the LIVAS database (Amiridis
et al., 2015). Therefore, the developed MIDAS dataset expands, complements and upgrades existing
observational capabilities of dust aerosols being suitable for research studies related to climatology,
model evaluation and data assimilation.
For the development of the fine resolution MIDAS DOD, a synergy of MODIS-Aqua (Section
2.1), MERRA-2 (Section 2.2), CALIPSO-CALIOP (Section 2.3) and AERONET (Section 2.4)
aerosol products has been deployed by taking advantage of the strong capabilities of each dataset.
Based on the applied methodology (Section 3.1), the DOD is calculated by the product of MODIS-
Aqua Level 2 AOD and the collocated DOD-to-AOD ratio from MERRA-2. The uncertainty of the
DOD is calculated from the uncertainties of both MODIS AOD and the MERRA-2 dust fraction,
using AERONET and CALIOP, respectively, as a reference (Section 3.2). We thoroughly compare
the MERRA-2 dust fraction against the CALIOP dust portion in Section 4.2. The MIDAS DOD is
evaluated against AERONET in Section 4.3 and compared with MERRA-2 and CALIOP DODs in
Section 4.4. In section 4.5 we provide the annual and seasonal global geographical distributions of
DOD. Finally, the main findings are summarized and are drawn in Section 5.



## 2. Datasets

### 2.1. MODIS

The MODerate resolution Imaging Spectroradiometer (MODIS) is a passive sensor measuring the top of atmosphere (TOA) reflectance in order to retrieve aerosol optical depth (AOD), among other aerosol optical properties, at various wavelengths spanning from the visible to the near infrared spectrum range. MODIS is mounted on the NASA's twin polar satellites Terra and Aqua acquiring high-quality aerosol data since 2000 and 2002, respectively, while thanks to its wide swath (~2330 km) provides near-global observations, almost on a daily basis. The derivation of AOD is achieved through the implementation of three retrieval algorithms based on the Dark Target (DT) approach, valid over oceans (Remer et al., 2002; 2005; 2008) and vegetated continental areas (Levy et al., 2007a; 2007b; 2010), or the Deep Blue (DB) approach (Hsu et al., 2004; Sayer et al., 2013) over land surfaces characterized by high reflectivity. Depending on the version of the retrieval algorithms, the MODIS datasets are organized at various collections as well as at various levels corresponding to their spatial and temporal resolution. For our purposes, we are utilizing the Collection 6 (C006) MODIS-Aqua Level 2 (L2) retrievals, over the period 2007-2016, which are reported at 5-min swath granules (Levy et al., 2013) and are accessible from the Level-1 and Atmosphere Archive & Distribution System (LAADS) Distributed Active Archive Center (DAAC) (https://ladsweb.modaps.eosdis.nasa.gov/). Each swath is composed by 203 x 135 retrievals, of increasing spatial resolution from the nadir view (10 km x 10 km) towards the edge of the satellite scan (48 km x 20 km), in which a Quality Assurance (QA) flag is assigned (Hubanks, 2018). More specifically, these bit values represent the reliability of the algorithm output and are equal to 0 ("No Confidence"), 1 ("Marginal"), 2 ("Good") and 3 ("Very Good"). As it has been mentioned above, the MODIS AOD retrievals are acquired based on different algorithms according to the underlying surface type. In order to fill observational gaps, attributed to the assumptions or limitations of the applied MODIS algorithms, the DT-Ocean (QA≥1), DT-Land (QA=3) and DB-Land (QA≥2) AOD retrievals are merged based on the Normalized Difference Vegetation Index (NDVI) and the highest accuracy criterion, as it has been presented by Sayer et al (2014). From the raw MODIS files, this "merged" AOD stored in the scientific data set (SDS) and named "AOD_550_Dark_Target_Deep_Blue_Combined" is extracted and processed for the needs of the current work. Finally, two quality filtering criteria are applied to the raw MODIS AODs for eliminating observations which may be unreliable. AODs associated with cloud fraction (CF) higher than 0.8 as well as those with no adjacent retrievals are masked out following the recommendations of previous studies (Anderson et al., 2005; Zhang and Reid, 2006; Hyer et al., 2011; Shi et al., 2011). The first criterion is associated with the potential cloud contamination of AODs, and the second attempts at removing "suspicious" retrievals from the dataset.



170

*2.2. MERRA-2*

172

The Modern-Era Retrospective Analysis for Research and Applications, version 2 (MERRA-2), developed by the NASA Global Modeling and Assimilation Office (GMAO), is the first atmospheric reanalysis spanning over the new modern satellite era (1980 onward) in which aerosol-radiation interactions and the two-way feedbacks with atmospheric processes are taken into account (Gelaro et al., 2017). The key components of MERRA-2 (Buchard et al., 2017) are the Goddard Earth Observing System (GEOS-5) (Rienecker et al. 2008; Molod et al. 2015), which is radiatively coupled to the Goddard Chemistry Aerosol Radiation and Transport model (GOCART; Chin et al. 2002; Colarco et al. 2010), and the three-dimensional variational data assimilation (3DVar) Gridpoint Statistical Interpolation analysis system (GSI) (Wu et al. 2002).

In the GOCART aerosol module, emission, sinks, removal mechanisms (dry deposition and gravitational settling, large-scale wet removal and convective scavenging) as well as the chemical processes of five aerosol species (dust, sea-salt, sulfate, and black and organic carbon) are simulated. Their optical properties are based on the updated Optical Properties of Aerosols and Clouds (OPAC) database (Hess et al. 1998), incorporating dust non-spherical shape (Meng et al. 2010; Colarco et al. 2014), and are calculated according to Colarco et al. (2010). For coarse particles (i.e., dust and sea-salt), five non-interacting size bins are considered whose emissions are driven by the wind speed based on the parameterizations of Marticorena and Bergametti (1995) for dust and the modified version of Gong (2003) for sea-salt. Both hydrophobic and hydrophilic black (BC) and organic (OC) carbon emitted from anthropogenic activities (i.e., fossil fuel combustion) and natural processes (i.e., biomass burning) are considered. Regarding sulfate aerosols ($SO_4$), these either are primarily emitted or are formed by the chemical oxidation of sulfur dioxide gas ($SO_2$) and dimethyl sulfide (DMS). Until 2010, daily emissions of eruptive and degassing volcanoes are derived from the AeroCom Phase II project (Diehl et al. 2012; http://aerocom.met.no/) and afterwards only a repeating annual cycle of degassing volcanoes is included in MERRA-2. The hygroscopic growth of sea-salt, sulfate and hydrophilic carbonaceous aerosols is determined by the simulated relative humidity (RH) and the subsequent modification of particles' shape and composition is taken into account in computations of particles' fall velocity and optical parameters (Randles et al., 2017). A detailed description of the emission inventories along with the global climatological maps, representative for the period 2000 – 2014, are given in Randles et al. (2017).

MERRA-2 is a multidecadal reanalysis in which a variety of meteorological and aerosol observations are jointly assimilated (Gelaro et al., 2017). The former group of observations consists of ground-based and spaceborne atmospheric measurements/retrievals summarized in Table 1 of





Gelaro et al. (2017) while the full description is presented in McCarty et al. (2016). For aerosol data
assimilation, the core of the utilized satellite data is coming from the MODIS instrument. Over
oceans, are also used AVHRR radiances, from January 1980 to August 2002, and over bright surfaces
(albedo > 0.15) the non-bias-corrected AOD (February 2000 – June 2014) retrieved for the Multiangle
Imaging SpectroRadiometer (MISR; Kahn et al., 2005) is assimilated. Apart from satellite datasets,
the Level 2 (L2) quality-assured AERONET retrievals (1999 – October 2014; Holben et al., 1998)
are integrated in the MERRA-2 assimilation system (Goddard Aerosol Assimilation System, GAAS)
which is presented in Randles et al. (2017; Section 3). From MODIS (above dark target continental
and maritime areas, Collection 5) and AVHRR (above oceanic regions), the AODs are retrieved from
the cloud-free radiances and adjusted (bias correction) to the corresponding AERONET AODs, via a
neural net retrieval (NNR). It must be clarified, that only the MERRA-2 AOD is directly constrained
by the observations while the model's performance (background forecast) and data assimilation
structure (parameterization of error covariances) are "responsible" for the aerosol speciation among
other aerosol diagnostics (Buchard et al., 2017).

In the present study, we use the columnar MERRA-2 total and dust AOD at 550 nm in order to

calculate the contribution, in optical terms, of mineral particles to the overall load. The computed
dust-to-total AOD ratio is evaluated against CALIOP retrievals and then used for the derivation of
dust optical depth (DOD) on MODIS-Aqua swaths. MERRA-2 products (M2T1NXAER files;
V5.12.4; aerosol diagnostics) have been downloaded from the GES DISC server
(https://disc.gsfc.nasa.gov/)) and are reported as hourly averages at 0.5° x 0.625° lat-lon spatial
resolution.

*2.3. CALIOP*

The Cloud Aerosol Lidar with Orthogonal Polarization (CALIOP), onboard the Cloud-Aerosol

Lidar and Infrared Pathfinder Satellite Observation (CALIPSO) satellite, provides altitude resolved
observations of aerosols and clouds since mid-June 2006 (Winker et al., 2010). CALIPSO, flying in
the A-Train constellation (Stephens et al., 2002), provides almost simultaneous observations with
Aqua thus making feasible and powerful their synergistic implementation for aerosol research.
CALIOP, an elastic backscatter two-wavelength polarization-sensitive Nd:YAG lidar in a near-nadir-
viewing geometry (since November 28, 2007, 3 degrees off-nadir), emits linearly polarized light at
532 and 1064 nm and detects the co-polar components at 532 and 1064 nm and the cross-polar
component at 532 nm, relative to the laser polarization plane (Hunt et al., 2009). Based on the
attenuated backscatter profiles (Level 1B) and the implementation of retrieval algorithms (Winker et
al., 2009), aerosol/cloud profiles as well as layer products are provided at various processing levels



(Tackett et al., 2018). CALIOP Level 2 (L2) aerosol and cloud products both are provided at a uniform
spatial resolution at horizontal (5 km) and vertical (60 m) dimensions. Detectable atmospheric
features are first categorized to aerosols or clouds and then are further discriminated at specific
subtypes according to Vaughan et al. (2009). For aerosols, in the Version 3 used here, 6 subtypes are
considered consisting of clean marine, dust, polluted continental, clean continental, polluted dust and
smoke (Omar et al., 2009). Based on the aerosol subtype classification, specific extinction-to-
backscatter ratios (Lidar Ratio - LR) are applied for the provision of extinction coefficient profiles
along the CALIPSO orbit-track (Young and Vaughan, 2009).

In this study we use the CALIOP pure dust product developed in the framework of the ESA-

LIVAS (Amiridis et al., 2015) database (http://lidar.space.noa.gr:8080/livas/) according to the
methodology described in Amiridis et al. (2013) and updated in Marinou et al. (2017). The
aforementioned technique relies on the incorporation of aerosol backscatter coefficient profiles and
depolarization ratio, providing a strong signal of dust presence due to mineral particles' irregular
shape (Freudenthaler et al., 2009; Burton et al., 2015; Mamouri and Ansmann, 2017), allowing the
separation of dust component from aerosol mixtures. For our purposes, instead of the raw universal
CALIOP dust LR (40 sr; Version 3), we are applying appropriate regionally-dependent LR values
(see Figure S1; Marinou et al., 2017; Proestakis et al. 2018 and references within), which are
multiplied with the dust backscatter coefficient profiles at 532 nm in order to calculate the
corresponding extinction coefficient profiles. After a series of strict quality screening filters (Marinou
et al., 2017), the columnar total/dust/non-dust optical depths as well as the DOD-to-AOD ratio over
the period 2007 – 2015 are aggregated at 1° x 1° grid cells covering the whole globe. The performance
of the pure DOD product has been assessed against AERONET over N. Africa and Europe (Amiridis
et al., 2013) revealing a substantial improvement when the abovementioned methodological steps are
applied. This has led to a broadening of research studies, such as the assessment of dust outbreaks
(Kosmopoulos et al., 2017; Solomos et al., 2018) and phytoplankton growth (Li et al., 2018), the 4D
description of mineral loads over long-term periods (Marinou et al., 2017; Proestakis et al., 2018),
the evaluation of dust models (Tsikerdekis et al., 2017; Georgoulias et al., 2018; Konsta et al., 2018)
and the evaluation of new satellite products (Georgoulias et al., 2016) in which the LIVAS pure DOD
product can be utilized.

*2.4. AERONET*

Ground-based observations acquired from the AEronet RObotic NETwork (AERONET; Holben

et al., 1998) have been used as reference in order to evaluate the accuracy of the quality assured





MODIS AOD as well as of the derived MODIS DOD product. The evaluation analysis has been
performed by utilizing the almucantar (inversion) retrievals, providing information for the total
aerosol amount (AOD) as well as for other microphysical (e.g., volume size distribution) and optical
(e.g., single scattering albedo) properties (Dubovik and King, 2000; Dubovik et al., 2006). In the
present study, focus is put on the aerosol optical properties retrieved at four wavelengths (440, 675,
870 and 1020 nm) utilizing as inputs spectral AODs and sky (diffuse) radiances. More specifically,
we used the Version 2 (V2) AERONET data of AOD (for total and coarse aerosols), Ångström
exponent ($\alpha$) and single scattering albedo (SSA). For the amount (AOD) and size ($\alpha$) related optical
parameters, the quality assured retrievals (i.e., Level 2; L2) are only used whereas for the SSA, the
L2 and Level 1.5 (L1.5) observations are merged in order to ensure maximum availability.
Unfavourable atmospheric conditions or restrictions on solar geometry result in a reduced amount of
inversion outputs with respect to the sun-direct measurements or the Spectral Deconvolution
Algorithm (SDA; O'Neill et al., 2003) retrievals. Even though both types of AERONET data provide
information about aerosol size (i.e., Ångström exponent) or coarse AOD (i.e., SDA), the optimum
approach for identifying dust particles and discriminating them from other coarse particles (i.e., sea-
salt) requires the of SSA, along with size optical properties, as it will be discussed in the next
paragraph.
Through the combination of the selected optical properties we achieved the spectral matching
between ground-based and spaceborne observations as well as the determination of DOD on
AERONET retrievals. Regarding the first part, the $\alpha_{440-870nm}$ and $AOD_{870nm}$ values are applied in the
Ångström formula in order to interpolate the AERONET AOD at a common wavelength (i.e., 550
nm) with MODIS. In contrast to the MODIS-AERONET AOD comparison, the corresponding
evaluation for DOD requires a special treatment of AERONET retrievals in order to define, as much
as possible based on columnar data, conditions where dust particles either only exist or clearly
dominate over other aerosol species. The vast majority of previous studies (e.g., Fotiadi et al., 2006;
Toledano et al., 2007; Basart et al., 2009) have relied on combining AOD and $\alpha$ for aerosol
characterization, associating the presence of mineral particles with low alpha levels and considerable
AODs. Here, we are keeping records where the $\alpha_{440-870nm} \leq 0.75$ and $SSA_{675nm} - SSA_{440nm} > 0$ without
taking into account the aerosol optical depth. The first criterion ensures the predominance of coarse
aerosols while the second one serves as an additional filter for discriminating dust from sea-salt
particles, taking advantage of the specific spectral signature of SSA (i.e., decreasing absorptivity for
increasing wavelengths in the visible spectrum) in pure or rich dust environments (Giles et al., 2012).
Then, from the coarse AODs at 440, 675 and 870 nm we calculate the corresponding $\alpha$, which is
applied in order to obtain the AERONET coarse AOD at 550 nm. This constitutes the AERONET-
derived DOD assuming that the contribution of fine dust particles (particles with radii less than the



inflection point in the volume size distribution) is small. Likewise, through this consideration any
potential "contamination" from small-size particles of anthropogenic or natural origin (e.g., biomass
burning), which is likely far away from the sources, is tempered or avoided.

**3. Methods**
*3.1. Derivation of dust optical depth on MODIS swaths*

The core concept of our approach is to derive DOD on MODIS L2 retrievals, provided at fine
spatial resolution, via the synergy with the MERRA-2 products. More specifically, the MERRA-2
dust fraction (MDF) to total $AOD_{550nm}$ (Eq. 1) is multiplied with the MODIS $AOD_{550nm}$ in order to
calculate $DOD_{550nm}$ at swath-level (Eq. 2).

$$MDF = \frac{AOD_{DUST;MERRA}}{AOD_{TOTAL;MERRA}} \text{ (Eq. 1)}$$

$$DOD_{MODIS} = AOD_{MODIS} * MDF \text{ (Eq. 2)}$$

To achieve that, the datasets are collocated temporally and spatially. MERRA-2 outputs are
provided at coarse spatial resolution (0.5° x 0.625°) in contrast to MODIS-Aqua observations (10 km
x 10 km). MODIS swaths are composed by 203 x 135 retrievals and for each one of them we compute
the distance from the MERRA-2 grid points, considering the closest hourly time step to MODIS
overpass time. Then, the MERRA-2 dust portion is used to calculate the DOD from the AOD on
MODIS swath native grid. Our approach avoids on purpose the inclusion of additional optical
properties providing information on aerosol size ($\alpha$) available from MODIS and absorptivity (Aerosol
Index) from OMI that are characterized by inherent limitations. Previous evaluation studies (Levy et
al., 2013) have shown that size parameters acquired by MODIS are highly uncertain, particularly over
land. In addition, since early 2008, the OMI sensor has lost half of its swath due to the "row-anomaly"
issue (Torres et al., 2018) thus "hampering" the MODIS-OMI collocation when it is attempted at fine
spatial resolution.

*3.2 Uncertainty estimation*

As expressed in Eq. 2, the MIDAS DOD results from the product of MODIS AOD and MDF from
MERRA-2. The uncertainty of the DOD product ($\Delta$(DOD)) accounts for the corresponding
uncertainties of the AOD and the MDF, which are calculated using AERONET and CALIOP,



respectively, as a reference. The mathematical expression of the Δ(DOD), given in Eq. 3, results from
the implementation of the product rule on Eq. 2.

$\Delta(DOD) = \Delta(AOD) * MDF + AOD * \Delta(MDF)$ (**Eq. 3**)

The term Δ(AOD) in Eq. 3, representing the expected error (EE) confidence envelope in which

~68% of the MODIS-AERONET AOD differences fall within, varies depending on the MODIS
aerosol retrieval algorithm applied. MODIS provides AODs above oceans, dark vegetated land areas
as well as over surfaces with high reflectivity (excluding snow- and ice-covered regions) based on
retrieval techniques relying on different assumptions whereas over transition zones between arid and
vegetated continental parts, DT and DB AODs are merged (Sayer et al., 2014).

For each retrieval algorithm, we use the linear equations expressing Δ(AOD) with respect to

AERONET AOD documented in the literature, for DT-Ocean (Levy et al., 2013; Eq. 4), DT-Land
(Levy et al., 2010; Eq. 5), and DB-Land (Sayer et al., 2013; Eq. 6) AODs. For the merged (DB+DT)
land AOD, the error is calculated via the square root of the quadrature sum of the DT-Land and DB-
Land uncertainties (Eq. 7). Before proceeding with the calculation of the Δ(DOD), few key aspects
must be highlighted for the sake of clarity. In equations 4 and 5, the AOD uncertainty is defined as a
diagnostic error since it is calculated utilizing AERONET as reference. Here, we are using the same
equations replacing AERONET AODs with those given by MODIS. This relies on the fact (results
not shown here) that their averages are almost unbiased. For the ocean AOD uncertainty, the defined
EE margins (Levy et al., 2013) have been modified in order to sustain symmetry by keeping the upper
bound (i.e., thus including more than 68% of MODIS-AERONET pairs within the EE). Sayer et al.
(2013) estimated the uncertainty of DB AOD by taking into account the geometric air mass factor
(AMF) resulting from the sum of the reciprocal cosines of the solar and viewing zenith angles (Eq.

6).


$\Delta(AOD_{DT-Ocean}) = \pm(0.10 * AOD + 0.04)$ (**Eq. 4**)

$\Delta(AOD_{DT-Land}) = \pm(0.15 * AOD + 0.05)$ (**Eq. 5**)

$\Delta(AOD_{DB-Land}) = \pm(\frac{0.086 + 0.56 * AOD}{AMF})$ (**Eq. 6**)

$\Delta(AOD_{DTDB-Land}) = \pm\sqrt{[\Delta(AOD_{DT-Land})]^2 + [\Delta(AOD_{DB-Land})]^2}$ (**Eq. 7**)


The CALIOP DOD-to-AOD ratio is our reference for estimating the uncertainty limits of the

MERRA-2 dust fraction (MDF). The analysis is performed at 1° x 1° spatial resolution considering
only grid cells in which both MERRA-2 and CALIOP DODs are higher or equal than 0.02. According



to this criterion, more than 450000 CALIOP-MERRA2 collocated pairs have been found which are
sorted (ascending order) based on MERRA-2 MDF (ranging from 0 to 1) and then are grouped in
equal size bins containing 20000 data each sub-sample. For every group, we computed the median
MDF (x axis) as well as the $68^{th}$ percentile of the absolute MERRA-2 – CALIOP dust fraction (y
axis) and then we found the best polynomial fit (Eq. 8).

$\Delta(MDF) = \pm(2.282 * MDF^4 - 6.222 * MDF^3 + 4.700 * MDF^2 - 0.969 * MDF + 0.199)$ (**Eq.**
**8**)
Depending on the selected MODIS algorithm, the appropriate combination between AOD (Eqs.
4, 5, 6 and 7) and MDF (Eq. 8) uncertainties is applied to calculate the $\Delta(DOD)$ (Eq. 3) on each
measurement (i.e., DOD), at each grid cell, throughout the study period. When averaging each grid
cell at each considered timescale the uncertainty is obtained by propagating each individual
measurement uncertainty, i.e., taking the square root of the sum of the quadratic $\Delta(DOD)$ divided by
the number of available measurements. We also estimate the uncertainty of the average due to
sampling using the standard error (i.e., the standard deviation divided by the square root of the number
of measurements). These two uncertainty quantities are in turn combined into a total uncertainty that
is calculated as the square root of their quadratic sum. The obtained findings will be discussed in
parallel with the global spatial patterns (Section 4.5) of dust optical depth in order to provide a
measure of the reliability of the derived DOD product.

**4. Results**

On the following sections, a series of analyses including an intercomparison between MERRA-2
and MODIS AODs (Section 4.1), the evaluation of MDF with respect to CALIOP (Section 4.2), an
evaluation of MIDAS DOD versus AERONET observations (Section 4.3) as well as an
intercomparison among MIDAS, CALIOP and MERRA-2 DODs (Section 4.4), is presented. All the
aforementioned steps are necessary in order to justify the validity of the applied methodology and to
understand its limitations. In the last section (4.5), the global annual and seasonal DOD patterns are
presented as a demonstration of the MIDAS dataset and the obtained spatiotemporal features are
briefly discussed since a climatological study it is the scientific topic of the companion paper.

*4.1. Intercomparison of MERRA-2 and MODIS AODs*

A prior step of our analysis is to investigate the consistency between MODIS-Aqua and MERRA-
2 AODs in order to ensure that they are not similar or identical thus making meaningless the





implementation of MODIS, which is not providing observations of high sampling frequency (single
overpasses) with respect to MERRA-2 products (hourly outputs). For their intercomparison, the
satellite and reanalysis datasets were regridded at 1° x 1° spatial resolution and they have been
temporally (satellite overpass – timestep) and spatially (grid cell coordinates) collocated. The average
global geographical distributions (2007-2016) of the simulated (MERRA-2) and retrieved (MODIS)
AODs are illustrated in Figures 1-i and 1-ii, respectively, along with their difference map (Figure 1-
iii) and the corresponding frequency histogram of MERRA-2 – MODIS AOD deviations (Figure 1-
iv).

A visual comparison between the two patterns does not reveal substantial deviations, in terms of

spatial characteristics, as indicated by the reproduction of the maximum AODs over major dust (i.e.
Sahara), biomass (i.e., Central Africa) and pollution (i.e., E. Asia) sources as well as over areas (i.e.,
Sub-Sahel) where very high concentrations of aerosol mixtures are recorded. A good agreement is
also apparent over downwind oceanic regions affected by short-to-long range transport of dust
(Tropical Atlantic Ocean) or biomass burning (South Atlantic Ocean). However, the difference map
(Fig. 1-iii) reveals substantial deviations, particularly over areas in which specific aerosol types
dominate. Across N. Africa, the simulated AODs (MERRA-2) are higher (reddish colors) than the
observed ones (MODIS), by up to 0.20-0.25, while positive MERRA-MODIS differences are also
encountered over other dust abundant areas such as the Taklamakan Desert and the southwestern
parts of Asia. On the contrary, higher MODIS than MERRA-2 AODs (bluish colors in Fig. 1-iii) are
predominant in central Africa (by up to 0.3) and evident in the Amazon basin (by up to 0.1) as well
as in Indonesia. In the latter regions, the columnar aerosol load mainly consists of carbonaceous
particles originating from agricultural burning and wildfires taking place from May to October in
central Africa (Bond et al., 2013), from landscape fires in the Amazon, with peak activity in August
and September (van der Werf et al., 2006) and from extended burned areas (Giglio et al., 2013) in
Indonesia, between August and October (Randerson et al., 2012). Negative differences (i.e. higher
MODIS AODs) also appear across the Gangetic Basin, where the major black carbon sources of India
are located (Paliwal et al., 2016), while the maximum values (exceeding 0.3) are recorded in the
heavily populated and industrialized regions of E. Asia (Zhang et al., 2015), emitting absorbing (black
carbon) and scattering (sulphate) fine pollution particles. Above oceans, the majority of reanalysis-
satellite departures are negligible, except over the tropical and southern Atlantic Ocean affected by
dust and biomass aerosol transport, respectively, with negative biases (i.e., lower MERRA-2 AODs)
hardly exceeding 0.05 in absolute terms.

The underestimated AODs in MERRA-2 over the major sources of biomass aerosols are probably

due to the negative increment in the assimilation system (Buchard et al., 2017). In Asia, GOCART
as well as the majority of the existing models underestimate the amount of BC aerosols produced by



man-made activities (Koch et al., 2009). Moreover, the anthropogenic OC/BC and $SO_2$ emissions
vary on yearly basis but in MERRA-2 are kept the same after 2006 and 2008, respectively (Randles
et al., 2017), while there is a lack of nitrate aerosols (Buchard et al., 2017). All the aforementioned
reasons could account for the underestimation of MERRA-2 AOD in Asia as already shown by
comparisons against AERONET measurements (Sun et al., 2019). Furthermore, the described
inherent weaknesses are intensified by the paucity of ground-based AERONET and cloud-free
MODIS retrievals thus reducing the availability of assimilated observations. In such cases, as it has
been mentioned by Buchard et al. (2017), the performance of MERRA-2 is driven by the underlying
forecast model (GEOS-5). Over N. Africa, as well as in other dust rich environments, the positive
reanalysis-satellite biases are mainly linked with the overestimation of GOCART dust aerosol
amounts, with respect to a variety of spaceborne observations, as it has been discussed by Yu et al.
(2010) and Chin et al. (2014). Moreover, over bright surfaces MERRA-2 assimilates uncorrected
MISR AODs, which are higher than the corresponding MODIS retrievals, particularly at low AOD
conditions, as it has been shown by Banks et al. (2013) and Farahat (2019). From a global and long-
term perspective, the positive and negative deviations of an enormous number of pairs span from -
0.3 to 0.3 and are almost equally separated around zero resulting in a Gaussian frequency distribution
(Fig. 1-iv).

*4.2. Evaluation of MERRA-2 dust portion versus CALIOP retrievals*

The evaluation of the MERRA-2 dust portion (i.e., MDF) is a critical step of our analysis since it
is used as the scaling factor of the MODIS AOD for the derivation of DOD. For this reason, the
corresponding columnar parameter computed from the quality assured and updated CALIOP profiles
(see Section 2.3) is used as reference. It must be highlighted that the only existing evaluation studies
of MERRA-2 aerosol products have been performed either for specific aerosol species or limited time
periods (Buchard et al., 2017; Veselovskii et al., 2018) showing the ability of MERRA-2 in
reproducing the integrated aerosol fields. Nevertheless, the speciation of the suspended particles,
which is to a large extent determined by the model physics assumptions (Gelaro et al., 2017), has not
been thoroughly evaluated. Therefore, the present analysis will complement and expand further the
current works providing insight about the performance of MERRA-2 in terms of discriminating
among aerosol types (particularly for dust) and subsequently estimating their contribution to the total
atmospheric load.
Figure 2 depicts the geographical distributions of the dust-to-total AOD ratio, based on MERRA-
2 (i) and CALIOP (ii), averaged over the period 2007 – 2015. The corresponding maps of mean bias,
fractional bias (FB), fractional gross error (FGE) and correlation coefficient (R) are given in Figure





3. For consistency, we regridded the MERRA-2 data to 1° x 1° spatial resolution and selected the
closest output to the CALIOP overpass time. Both datasets provide nighttime observations; however,
the analysis has been restricted to sunlight hours only when aerosol retrievals obtained by passive
sensors at visible wavelengths are assimilated. At a first glance, the spatial patterns are very similar,
particularly in areas where the presence of dust is predominant. Across the "dust belt" (Prospero et
al., 2002), the most evident deviations (underestimation of MERRA-2 dust portion by ~0.1 or 10%)
are recorded in the borders of Afghanistan and Pakistan (Dasht-e Margo and Kharan Deserts) as well
as in the Taklamakan Desert (Fig. 3-i). However, from the FB (Fig. 3-ii) and FGE (Fig. 3-iii) maps it
is evident that the calculated values in most of the aforementioned regions are close to zero (ideal
score) thus indicating a very good performance of MERRA-2. In terms of temporal covariation (Fig.
3-iv), moderate R values (~0.5) are computed while in the western parts of Sahara the correlation
levels are slightly higher than zero. Due to the complex and highly variable nature of the emission
processes and therefore the poorer behavior of the model, the correlation tends to be smaller over the
main dust sources. In downwind regions of the N. Hemisphere, particularly over the main transport
paths (i.e., Atlantic Ocean, Mediterranean, Arabian Sea, E. Asia), correlation substantially increases
(up to 0.9). This is further supported by the FB and FGE metrics; which, however, downgrade for
increasing distances from the sources due to the reduction of dust contribution to the total aerosol
load. An exception is observed for the mean bias along the tropical Atlantic Ocean where the
MERRA-2 dust portion is overestimated by up to 10% in its eastern parts in contrast to longitudes
westward of 45° W where zero biases or slight underestimations (~5%, Caribbean Sea) are computed.
A discrepancy between the CALIOP and MERRA-2 dust portion is found in the Mojave, Sonoran,
Chihuahuan desert areas extending between southwestern US and northern Mexico. As shown in
Figure 2, the dust contribution in those areas is more widespread and stronger based on spaceborne
retrievals in contrast to MERRA-2, which simulates less dust amount over the sources (Mojave
Desert) and the surrounding regions. According to the evaluation metrics (Figure 3), the
underestimation of MERRA-2 dust contribution to the total aerosol load ranges between 20% to 50%,
negative FB (down to -1) and high FGE values (locally exceeding 1) are evident while the correlation
levels are low, particularly over Mexico. In the Southern Hemisphere, the deficiency of MERRA-2
is pronounced along the western coasts of S. America as well as in the Patagonian and Monte Deserts,
both situated in Argentina. Similar results are found in South Africa while in Australia a contrast
between its western/eastern and central parts with slight MERRA-2 underestimations and
overestimations, up to 20% in absolute terms, respectively, are recorded (Figure 3-i). Nevertheless,
the agreement between MERRA-2 and CALIOP in temporal terms is supported by the moderate-to-
high R values over the "hotspot" regions (Figure 3-iv). Outside of the main dust-affected regions, an
obvious discrepancy is found in the eastern Canada and northeastern Russia where MERRA-2 dust



contribution yields very low values (< 20%) in contrast to CALIOP reaching up to 50%. Due to their
geographical position, the occurrence of dust loads might not be frequent there but their contribution
to the total load can be significant under low AOD conditions, mainly recorded in the region. This
indicates a poor representation by MERRA-2; however, it must also be taken into account a potential
cloud contamination in the lidar signals.
The discrepancies are mainly driven by the partial representation of dust sources in MERRA-2
resulting in potentially underestimated dust emission areas and subsequently to lower dust
contribution to the total burden. In many areas dust is originated either from natural (arid lands, salt
lakes, glacial lakes) or from anthropogenic sources (Ginoux et al., 2012). Nevertheless, dust sources
in MERRA-2 are based on Ginoux et al. (2001) accounting mostly for natural dust emission areas.
This could partly explain the higher CALIOP dust contribution levels. Interestingly, most of the
positive CALIOP-MERRA-2 differences (i.e. bluish colors in Figure 3-i) are recorded in mountainous
areas characterized by complex terrain. Due to the variable geomorphology, the enhanced surface
returns contaminate the CALIOP signal close to the ground leading to higher columnar AODs and
lower contribution by dust loads suspended aloft. In addition, depending on the homogeneity of the
atmospheric scene over regions characterized by complex topography, variations in the optical paths
of subsequent CALIPSO L2 aerosol profiles considered in the L3 product may result to unrealistic
DOD and AOD values. Previous evaluation studies (e.g., Omar et al., 2013) have shown that CALIOP
underestimates AOD with respect to ground-based AERONET retrievals, particularly over desert
areas (Amiridis et al., 2015), which was attributed primarily to the incorrect assumption of the lidar
ratio (S) (Wandinger et al., 2010) and secondarily to the inability of the lidar to detect thin aerosol
layers (particularly during daytime conditions due to low signal-to-noise ratio). The former factor is
related to aerosol type and for Saharan dust particles the necessary increase of S (from 40 to 58 sr)
improved substantially the level of agreement versus AERONET and MODIS (Amiridis et al., 2013).
Similar adjustments (increments) on the raw S values, which are highly variable (Müller et al., 2007;
Baars et al., 2016), considered in the CALIOP retrieval algorithm have been applied in other source
areas of mineral particles (see Section 2.3; Figure S1). An additional factor that must be taken into
account, is the number of MERRA-2 – CALIOP pairs which is used for the metrics derivation. The
corresponding global geographical distribution (Figure S2-i), representative over the period 2007-
2015, shows that in areas where the model-satellite agreement is good (Figure 3) the number of
common samples is high (>100) in contrast to regions (<50) where the computed metrics are
degraded.
In order to complete the evaluation of the MERRA-2 dust portion versus CALIOP, the
dependency of the level agreement on the spatial representativeness within the 1°x1° CALIOP grid-
cell has also been investigated. Figure S2-ii displays the long-term averaged geographical distribution



of the number of CALIOP L2 profiles (up to 23) aggregated for the derivation of the 1°x1° grid-cell.
According to the global map, the maximum number of CALIOP profiles are recorded in the latitudinal
band extending from 45° S to 45° N while the "impact" of extended clouds around the equator is
apparent. Outside of this zone, the number of profiles used is mainly less than 14 and decreases
towards the poles due to the enhanced cloud coverage. The same evaluation metrics presented in
Figure 3 have been also computed at planetary scale for individual classes of CALIOP L2 number of
profiles (Figure S3). Overall, about 3.4 million pairs (x tick named "ALL" in Figure S3-a) have been
found over the period 2007-2015 and are almost equally distributed for bins spanning from 8 to 20
while the number of collocated data is higher in the lowermost ($\leq 7$) and uppermost ($\geq 21$) tails of the
distribution. Based on FB (Figure S3-c), FGE (Figure S3-d) and correlation (Figure S3-e) results it is
revealed that the consistency between MERRA-2 and CALIOP gradually improves for higher
CALIOP grid-cell representativeness. In quantitative terms, FB decreases from ~1.2 to ~0.2, FGE
decreases from ~1.6 to ~ 0.9 and R increases from ~0.5 to ~0.8 while the corresponding overall results
(i.e., first red bar) are equal to ~0.7, ~1.3 and ~0.7, respectively. At global scale, MERRA-2
overestimates dust portion by up to 1.5% with respect to CALIOP (Figure S3-b). Among the bin
classes, the maximum overestimation (~2.8%) is recorded when four CALIOP profiles are averaged
for the derivation of the 1°x1° cell, while the positive MERRA-2-CALIOP differences become lower
than 0.1 when at least 12 CALIOP profiles are considered. Regarding the bias sign, the only exception
is observed for cases where 22 or 23 CALIOP profiles are used resulting in slight MERRA-2
underestimations.
Based on the findings of the Sections 4.1 and 4.2, the MERRA-2 related outputs (AOD, MDF)
used in this method, after their quantitative evaluation with MODIS and CALIPSO products, showed
reasonable results comparing them on a global scale. In order to provide a deeper assessment of the
derived DOD product we have used AERONET AOD and DOD data for an additional evaluation
purpose.

*4.3. Evaluation of MIDAS DOD versus AERONET observations*

In the present section, we provide an assessment of MODIS L2 AOD and the derived MIDAS
DOD against the corresponding AERONET almucantar retrievals as discussed in Section 2.4. The
validation of the MODIS quality filtered AOD (Section 2.1) aims at assessing the performance of the
input data while for the derived DOD to check the validity of our approach (Section 3.1). An
illustration of the MODIS-AERONET collocation is shown in Figure S4. At first, a short discussion
is made on the MODIS-AERONET AOD evaluation results, shown in Figure S5, obtained by 59445
pairs (Figure S5-ii) collected at 645 ground stations (Figure S5-i) during the period 2007-2016. Based
on the 2D histogram scatterplot of Figure S5, the very high correlation (R=0.894), the slope ($\alpha$=0.929)
which is close to unity and the near-zero offset ($b$=0.008) reveal a remarkable MODIS-AERONET
agreement. Overall, MODIS slightly underestimates AOD (-0.003 or -1.741% in absolute and relative
percentage terms, respectively) with respect to AERONET. Our results are consistent with those
obtained by dedicated global evaluation studies (Levy et al., 2013; Sayer et al., 2013; Sayer et al.,
2014) of C006 MODIS AOD product despite the differences regarding the time periods,
spatiotemporal collocation criteria, filtering of satellite retrievals based on QA flags and the
consideration of AERONET data.

The corresponding analysis for DOD is presented at global and station levels in Figures 4 and 5,

respectively. As expected, the coincident spaceborne and ground-based DODs collected at 376
AERONET stations (red circles in Figure 4-i) are drastically reduced down to 7299 pairs due to the
implementation of filters for the determination of DOD on AERONET data (Section 2.4). However,
the global scatterplot metrics (Figure 4-ii) are similar to those computed for AOD revealing a very
good performance of the MODIS derived DOD. Both datasets are well correlated (R=0.882) with
MIDAS slight underestimating DOD compared to AERONET (-0.009 or -5.307%). According to our
methodology, only the AERONET AODs associated with $\alpha$ lower/equal than/to 0.75 are kept for the
evaluation procedure. The defined upper threshold on $\alpha$ values is higher compared to previous
findings or applied cut-off levels (e.g. Dey et al., 2004; Tafuro et al., 2006; Reid et al., 2008; Kim et
al., 2011; Gkikas et al., 2016). We repeated the analysis by reducing $\alpha$ from 0.75 to 0.25 (results not
shown here) and obtained very similar global scatterplot metrics.

The evaluation analysis was also performed individually for each station. Figure 5 depicts only

sites with at least 30 common MODIS/AERONET observations, thus making the comparison at
station level meaningful. This criterion is satisfied in 61 stations, which overall comprise 70% (or
5085) of the total population of MODIS-AERONET coincident DODs, and are mostly located over
dust sources as well as on areas affected by dust transport, from short to long range. Figure 5-i shows
the station-by-station variability of the number of common MODIS/AERONET observations ranging
from 100 to 355 (Banizoumbou, Niger) across N. Africa and Middle East whereas in the remaining
sites is mainly lower than 70. Between the two datasets, very high R values (up to 0.977) are found
in N. Africa, Middle East, outflow regions (Cape Verde, Canary Islands, Mediterranean) and at distant
areas (Caribbean Sea) affected by long-range transport. Over the stations located across the Sahel,
the maximum RMSE levels (up to 0.243) are recorded (Figure 5-iii) due to the strong load and
variability of the Saharan dust plumes. The maximum positive biases (0.133) indicate that the derived
MIDAS DOD is overestimated. Several reasons may explain the predominance of positive MIDAS-
AERONET differences over the above-mentioned stations taken into account that the MERRA-2 dust
portion is adequately reproduced with respect to CALIOP. The first one is related to the MODIS



retrieval algorithm itself and more specifically with the applied aerosol models, surface reflectance
and cloud screening procedures (Sayer et al., 2013). However, for AOD (results not shown here),
small negative biases are observed in agreement with the findings by Sayer et al. (2013) who utilized
DB products only. The second factor is the absence of fine DOD on AERONET data which would
have reduced the obtained positive differences but its contribution to the total dust AOD it is difficult
and probably impossible to be quantified. Similar tendencies, but at a lesser degree, are found for the
RMSE and bias scores in the Middle East where the satellite and ground-based DODs are in general
well-correlated. In the Mediterranean, the temporal covariation between the two datasets is quite
consistent (R>0.8) with the MIDAS DOD being slightly underestimated probably due to the
underestimation of the MERRA-2 dust portion.
In Asia, few stations are available with sufficient number of MODIS-AERONET collocations in
which negative biases are generally recorded both for DOD (Figure 5-iv) and AOD (results not shown
here). This agreement indicates that the MODIS AOD underestimation is "transferred" also to DOD
and locally can be further enhanced by the MERRA-2 dust portion underestimation (Figure 3-i).
Along the western coasts of the United States, the evaluation scores at 5 AERONET sites show that
the performance of the retrieved AOD is superior than those for the derived DOD (R: 0.42-0.78, bias:
-0.02-0.05, RMSE: 0.03-0.07) attributed to the deficiency of MERRA-2 to reproduce adequately the
contribution of dust particles to the total dust aerosol load. Finally, our assessment analysis in the
Southern Hemisphere, for stations located in Argentina, Namibia and Australia, indicate slight
MODIS-AERONET deviations, spanning from -0.03 (Lucinda) to 0.02 (Gobabeb), and correlations
ranging from 0.12 (Fowlers_Gap) to 0.8 (Birdsville).

*4.4. Intercomparison of MIDAS, MERRA-2 and CALIOP DOD products*

Following the evaluation of MIDAS DOD against AERONET, the three DOD products derived
from MIDAS, MERRA-2 and CALIOP are investigated in parallel. For this purpose, the MERRA-2
outputs and MODIS retrievals have been regridded to 1° x 1° grid cells in order to match CALIOP's
spatial resolution while the study period extends from 2007 to 2015, driven again from CALIOP's
temporal availability. Then, the three datasets have been collocated by selecting the coincident pixel
for which the temporal deviation between model outputs and satellite overpasses is minimized. The
intercomparison has been performed only during daytime conditions and the obtained findings are
presented through geographical distributions (Section 4.4.1) and intra-annual timeseries of regional,
hemispherical and planetary averages (Section 4.4.2). Finally, it must be clarified that our focus in
this part of the analysis is the intercomparison of the different DOD products and not to interpret their





spatiotemporal features which will be discussed thoroughly in a companion paper analyzing the
DODs from the MIDAS fine resolution dataset.

4.4.1. *Geographical distributions*

The annual geographical distributions of CALIOP, MERRA-2 and MIDAS DODs are depicted
in Figures 6-i, 6-ii and 6-iii, respectively, while the corresponding global seasonal maps are provided
in Figure S6. Among the three datasets, both for annual and seasonal geographical distributions, it is
apparent a very good agreement in spatial terms in contrast to the magnitude of the simulated
(MERRA-2) and retrieved (MIDAS, CALIOP) DODs. The most evident deviations of MERRA-2
(Figure 6-ii) and MIDAS (Figure 6-iii), with respect to CALIOP (Figure 6-i), are encountered across
N. Africa forming clear patterns with positive and negative biases over the Sahara and the Sahel,
respectively. In particular, MERRA-2 DOD overestimations range from 0.04 to 0.20 while the
MIDAS-CALIOP deviations are lower; placing our DOD product between active remote sensing
retrievals and reanalysis dataset. A common feature is the location where the maximum
"overestimations" are observed. These areas are identified in Algeria, Niger and Chad featuring
substantially high dust concentrations. Previous studies relied on satellite (Yu et al., 2010; Kittaka et
al., 2011; Ma et al., 2013) and ground-based (Schuster et al., 2012; Omar et al., 2013) observations
have noted that CALIOP underestimates AOD over the Sahara. Konsta et al. (2018), who utilized
higher and more realistic dust lidar ratio (55 sr; adopted also for the region in the current study), with
respect to the aforementioned works (40 sr), reported similar tendencies against MODIS. Therefore,
other factors might contribute to the lower lidar-derived DODs over the arid regions of N. Africa. For
example, it has been observed that CALIOP can misclassify as clouds very intense dust layers which
can also attenuate significantly or totally the emitted laser beam (Yu et al., 2010; Konsta et al., 2018).
All these aspects, most likely met over dust sources, act towards reducing the extinction coefficient
and may explain the "missing" hotspot on CALIPSO in/around the Bodélé Depression in contrast to
single-view, multi-angle and geostationary passive satellite sensors (e.g., Banks and Bridley, 2013;
Wei et al., 2019). Across the Sahel, CALIOP provides higher DODs (mainly up to 0.2) both against
simulated and satellite products. These differences might be attributed to the misrepresentation of
dust sources in MERRA-2 along this zone where vegetation cover has a prominent seasonal cycle
(Kergoat et al., 2017). An inaccurate representation of vegetation also impacts the surface reflectance
which in turn can introduce critical errors in the retrieval algorithm. Sayer et al. (2013) showed that
MODIS overestimates AOD with respect to AERONET while the maximum MIDAS-AERONET
negative DOD bias (-0.154) is recorded at Ilorin (Figure 5-iv).



Over the eastern Tropical Atlantic Ocean, the difference between CALIOP and MIDAS is
negligible whereas MERRA-2 underestimates DOD by up to 0.08. In the Middle East, MERRA-2
and MIDAS DODs are higher than those retrieved by CALIOP over the Tigris-Euphrates basin while
an opposite tendency, particularly for MIDAS, is found in the interior parts of Saudi Arabia. Lower
DODs are also observed by CALIOP over the arid/semi-arid regions, including the Aral Sea,
eastwards of the Caspian Sea. This area of the planet is one of the most challenging for passive
observations from space due to the terrain complexity prohibiting the accurate characterization of the
surface reflectance and type. Such conditions impose artifacts to the retrieval algorithm resulting in
unrealistically high MODIS AODs (Klingmüller et al., 2016; see interactive comment posted by
Andrew Sayer) that may also affect MERRA-2 via assimilation. Along the mountainous western parts
of Iran, CALIPSO DOD is substantially higher than those derived with our methodology while
against MERRA-2 the obtained positive or negative differences are close to zero. The largest negative
MIDAS-CALIOP differences (exceeding 0.2), not only in Asia but all over the world, are recorded
in the Taklamakan Desert whereas the corresponding results between MERRA-2 and CALIOP are
somewhat lower. This might be attributed to an inappropriate selection (overestimation) of the lidar
ratio taking into account that CALIOP mainly underestimates AOD over the region, dust contribution
to the total AOD exceeds 70% (Proestakis et al., 2018), throughout the year, and MDF shows robust
consistency (Figure 3). Eastwards of the Asian continent, the situation is reversed and the CALIOP
DODs are lower by up to 0.2 when compared to MERRA-2 and MIDAS indicating a weaker trans-
Pacific transport, predominant during boreal spring (second row in Figure S6), being in agreement
with the findings of Yu et al. (2010) and Ma et al. (2013). In the S. Hemisphere, negative MERRA-
2-CALIOP and MIDAS-CALIOP differences are computed in Patagonia, which are not however
spatially coherent. On the contrary, in the desert areas of the inland parts of Australia, there is a clear
signal of positive MERRA-2-CALIOP deviations, not seen between MIDAS and CALIOP, most
likely attributed to the overestimation of aerosol (dust) optical depth by MERRA-2 as it has been
recently presented by Mukkavilli et al. (2019). On a global and long-term perspective, based on more
than 470000 collocated data, MERRA-2 correlates better with CALIOP than MIDAS (R=0.740 vs.
0.665), but is more biased (relative bias=4.264% vs. 9.405%).

*4.4.2. Planetary, hemispherical and regional intra-annual variability*
At a next step of the intercomparison, the variability of the planetary (Figure 7-i) and
hemispherical (Figures 7-ii, 7-iii) monthly averages of CALIOP (black curve), MERRA-2 (red curve)
and MIDAS (blue curve) DODs were compared. It is clarified that the calculations presented here
have been performed, at each considered timescale, following the upper branch shown in Figure 5 of
Levy et al. (2009), comprising first a temporal averaging and then a spatial averaging taking into the



weight of the grid cell surface area to the total domain area with available data. In the N. Hemisphere,
the most evident deviations among the three products occur from March to June when the dust activity
is more pronounced (Figure 7-ii). During the high-dust seasons, the DOD peaks are recorded in June,
being identical (0.117) for MERRA-2 and MIDAS with CALIOP giving lower levels (0.114). Both
the MIDAS/MERRA-2 "temporal consistency" and the CALIOP "underestimation" are mostly valid
during boreal spring and summer with few exceptions, highlighted when focus is given on the
temporal covariation. Within the course of the year, CALIOP and MERRA-2 DODs gradually
increase from January to June while decrease during the second half of the annual cycle. While the
trends in MIDAS DOD are similar overall, there is a local minimum observed in May (0.083)
resulting in deviations of -0.009 and -0.021 compared to CALIOP and MERRA-2, respectively. On
an annual basis (Table 1), the averaged MERRA-2 and MIDAS DODs, for the northern hemisphere,
are the same (0.055) and higher by up to 10% than the corresponding CALIOP mean (0.050). In the
S. Hemisphere (Figure 7-iii), DODs range at very low levels (up to 0.011), attributed to the low
amounts of mineral particles emitted from spatially restricted desert areas, and the limited dust
transport over oceanic regions. Despite these low values, there is an annual cycle pattern, which,
however, is not commonly reproduced by the three datasets in contrast to the annual means which are
almost identical among them (0.007-0.008; Table 1). In particular, MIDAS and MERRA-2 DODs are
maximized in February (0.009) while the highest levels for CALIOP are recorded in September
(0.011). For all DOD products, the minimum values (slightly less than 0.006) are found in May, which
are slightly lower than those observed during April-July (austral winter). At global scale (Figure 7-i),
the seasonal patterns of DODs are mainly driven by those of the N. Hemisphere, where the main dust
sources of the planet are situated, but the intra-annual cycles of MIDAS and MERRA-2 are not
identical with those of GLB in contrast to CALIOP. More specifically, there are two peaks (~0.05,
March and June) for MIDAS, flat maximum levels (~0.05) between March and June for MERRA-2
while there is a primary (0.048) and a secondary (0.041) maximum in June and March, respectively,
for CALIOP DODs. Even though there are month-by-month differences, the CALIOP (0.029),
MERRA-2 (0.031) and MIDAS (0.031) annual DODs are very close indicating a sufficient level of
agreement among the three datasets (Table 1). Likewise, our findings are almost identical with the
global DOD average (0.030) reported by Ridley et al. (2016).

The consistency among the three datasets, in terms of DOD magnitude and temporal covariation,

is highly dependent on the region of interest. Table 1 lists the computed annual averages as well as
their minimum/maximum limits, Figure S7 shows the defined sub-domains and Figure S8 their intra-
annual timeseries. The best agreement among MIDAS, CALIOP and MERRA-2 DODs is found along
the Tropical Atlantic Ocean, which is affected by Saharan dust transport throughout the year,
particularly its eastern sector. In the nearby outflow regions, considerably high DODs (> 0.1) are



found between January-August, being maximum in June, as indicated by the three datasets, with slight
underestimations in MERRA-2 (Fig. S8-k). Over the western Tropical Atlantic Ocean, the sharp
increase of DOD from May to June indicates the arrival of considerable amounts of Saharan particles,
which are sustained at high levels in summer and diminish during autumn and winter (Fig. S8-q).
This seasonal fluctuation is almost identically reproduced by the spaceborne (MIDAS, CALIOP) and
reanalysis (MERRA-2) products. Nevertheless, when the dust activity is well established in the area
(boreal summer), there is clear difference on DOD values with higher CALIOP levels compared to
MERRA-2 and MIDAS, which are quite similar.
Across N. Africa, and particularly in the Bodélé (Fig. S8-a) and W. Sahara (Fig. S8-h), the
CALIOP DODs are substantially lower when compared to MIDAS and MERRA-2. In the Bodélé,
this is evident for the entire year and in W. Sahara it can be clearly seen during the high-dust boreal
summer season. Similar findings are drawn either for other source areas, such as Central Asia (Fig.
S8-c), the northern Middle East (Fig. S8-d), the southwest United States (Fig. S8-e) or outflow
regions, such as the Mediterranean (Fig. S8-m). Over the Taklamakan (Fig. S8-f) and Gobi (Fig. S8-
b) Deserts, the CALIOP DODs are higher than the corresponding MIDAS and MERRA-2 regional
averages in April-May. Among the three DOD products, a very good temporal agreement it is found
in the Thar Desert (Fig. S8-g), but there are deviations regarding the peak of July which is higher in
MIDAS (1.172) than in CALIOP (0.978) and MERRA-2 (0.484), respectively. Over downwind
continental areas downwind of E. Asia (Fig. S8-i), only few exceptions break down the consistency
between MIDAS and CALIOP and MERRA-2 is able to reproduce the annual cycle but
underestimates the intensity of dust loads. In southern Middle East (Fig. S8-n), the reanalysis and the
spaceborne lidar DODs are very well correlated and reveal minor differences within the course of the
year. MIDAS captures satisfactorily the monthly variability of DOD, despite the local minimum in
May, but fails to reproduce the magnitude of the recorded maximum in June. Over the Northern
Pacific, Asian dust is transported eastwards during spring affecting nearby (Fig. S8-p) and distant
(Fig. S8-j) oceanic areas. The "signal" of this mechanism is clearly evident on MIDAS and MERRA-
2 timeseries in contrast to CALIOP which exhibits substantially lower DOD maxima. Moreover, they
appear earlier (March) with respect to the other two datasets (April) in the western North Pacific
Ocean. Based on MERRA-2 and MIDAS, in the sub-Sahel (Fig. S8-o), a primary and a secondary
maximum are recorded in March and October, in agreement with ground-based visibility records
(N'Tchayi Mbourou et al., 1997). CALIOP reproduces both peaks, but with a weaker intensity in
March compared to MIDAS and MERRA-2. However, throughout the year, the maximum CALIOP
DOD is observed in June (a local maximum is also recorded in MIDAS), which might be unrealistic
since it is not expected the accumulation of dust particles during summer months, when the
precipitation heights in the area are maximized. Saharan dust aerosols, under the impact of the





northeasterly harmattan winds, are carried over the Gulf of Guinea (Fig. S8-l) during boreal winter,
although DODs among the three datasets reveal a noticeable variability in terms of intensity.

*4.5. MIDAS DOD global climatology*

The annual and seasonal DOD patterns averaged over the period 2007 – 2016 are illustrated in
Figures 8 and 9. Among the desert areas of the planet, the most intense dust loads (DODs up to ~1.2;
Fig. 8) are hosted across the Sahara Desert and particularly in the Bodélé Depression of the northern
Lake Chad Basin (Washington et al., 2003). Over the region, these high DODs are sustained
throughout the year (Fig. 9) while due the prevailing meteorological conditions, during MAM (Fig.
9-ii) and JJA (Fig. 9-iii), mineral particles are transported westwards, along the Sahel, contributing
to the locally emitted anthropogenic dust (Ginoux et al., 2012). Substantial high climatological DODs
(up to 0.6; Fig. 8) are recorded in the western sector of Sahara, in contrast to the eastern parts,
attributed to the accumulation of dust aerosols primarily in JJA (Fig. 9-iii) and secondarily in MAM
(Fig. 9-ii), under the impact of the Saharan Heat Low (Schepanski et al., 2017). Saharan dust is
subjected to short-range transport affecting frequently the nearby maritime areas of the Gulf of
Guinea (Ben-Ami et al., 2009), the Mediterranean Sea (Gkikas et al., 2015) as well the Red Sea
(Banks et al., 2017). Nevertheless, the strongest signal of Saharan dust transport appears over the
Tropical Atlantic Ocean with massive loads of mineral particles, confined within the Saharan Air
Layer (SAL; Kanitz et al., 2014), reaching the Caribbean Sea (Prospero, 1999), under the impact of
the trade winds. The characteristics of the transatlantic dust transport reveal a remarkable intra-annual
variation (Fig. 9) as it concerns plumes' latitudinal position, longitudinal extension and intensity,
being maximum during boreal summer (Fig. 9-iii).
Dust activity over the Middle East is more pronounced in a "zone" extending from the alluvial
plain of the Tigris-Euphrates River to the southern parts of the Arabian Peninsula (Fig. 8), through
the eastern flat-lands of Saudi Arabia (Hamidi et al., 2013). Mineral particles emitted from these
sources affect also the Persian Gulf (Gianakopoulou and Toumi, 2011) and the Red Sea (Banks et al.,
2017); however, the major transport pattern is recorded across the northern Arabian Sea in JJA (Fig.
9-iii), when dust plumes can reach the western coasts of India (Ramaswamy et al., 2017). In the Asian
continent, the Taklamakan Desert (Ge et al., 2014), situated in the Tarim basin (NW China), consists
one of the strongest dust source of the planet yielding DODs up to 1 during spring (Fig. 9-ii). These
intensities are substantially higher than those recorded in the Gobi Desert, situated eastwards in the
same latitudinal band, due to the different composition of the erodible soils (Sun et al., 2013).
Midlatitude cyclones, propagating eastwards during springtime (Fig. 9-ii), mobilize dust emission
from both sources inducing uplifting and subsequently advection of mineral particles towards the



continental E. Asia (Yu et al., 2019) as well as over the north Pacific Ocean (Yu et al., 2008) and
exceptionally over the United States (Husar et al., 2001). Other hotspots of dust activity in Asia, are
recorded in the central parts (Li and Sokolik, 2018) and in the Sistan Basin (Alizadeh Choobari et al.,
2013). Dust aerosols originating from agricultural activities along the Indus River basin (Ginoux et
al., 2012) and natural processes in the Thar Desert (Proestakis et al., 2018) result in the accumulation
of mineral particles in the Pakistan-India borders while under favorable meteorological conditions
these loads are carried towards the Indo-Gangetic plain mainly during the pre-monsoon season (Dey
et al., 2004). In North America, dust production becomes more evident in the southwestern United
States and northwest Mexico in regional terms and during spring within the course of the year (Fig.
9-ii). However, DODs are mostly lower than 0.2, with few local exceedances, indicating weak dust
emission from the natural (Mojave and Sonoran Deserts; Hand et al., 2017) and anthropogenic
(Chihuahuan Desert and Southern Great Plains; Hand et al., 2016) dust sources of the region. Between
the two hemispheres, there is a clear contrast in DODs, being substantially lower in the S.
Hemisphere, attributed to the weaker processes triggering dust emission from the spatially restricted
deserts located in S. Africa (Bryant et al., 2007), S. America (Gassó and Torres, 2019) and in the
interior parts of Australia (Prospero et al., 2002).
Apart from the global climatological DOD pattern in Figure 8-i, the corresponding distributions
of the total uncertainty (taking into account the propagation of each individual DOD uncertainty to
the average grid cell value as well as the sampling uncertainty expressed by the standard error) and
the temporal availability are shown in Figs. 8-ii and -iii, respectively, thus allowing to assess the
accuracy of the derived product as well as its representativeness throughout the study period (2007-
2016). More than 70% of satellite retrievals, with respect to the full period, are participating in the
calculation of the mean DODs (Fig. 8-i) over the cloud-free desert areas while over dust-affected
downwind regions the corresponding percentages range from 30 to 60% (Fig. 8-iii). Regarding the
total DOD uncertainty (Fig. 8-ii), the spatial pattern indicates maximum absolute levels (up to 0.1) in
the sub-Sahel and in the Gulf of Guinea whereas similar values are found in the Taklamakan Desert
and lower in the Tropical Atlantic. On a seasonal basis (Figure S9), the spatial features of DOD
uncertainties are driven by those of DOD (Figure 9) over sources and receptor areas while MIDAS
DODs are similarly less reliable, in absolute terms, over regions affected rarely by dust loads.
However, in relative terms, across deserts and areas subjected to dust transport, the DOD uncertainty
with respect to the obtained long-term averages, throughout the year, is mainly lower than 20% and
over regions with weak dust loads is higher than 60%. This contrast is attributed to the larger number
of available DODs over/nearby the deserts, where dust signal is maximized, as well as to the lower
$\Delta$(DOD) resulting from the lower relative AOD errors and the better MDF performance there.



## 5. Summary and conclusions

In the current study, we presented the MIDAS (ModIs Dust AeroSol) dust optical depth (DOD) dataset, combining MODIS-Aqua AOD and DOD-to-AOD ratio extracted from collocated MERRA-2 reanalysis outputs. The derived fine resolution (0.1° x 0.1°) global dataset, valid for a decade (2007-2016), provides DOD both over continental and oceanic areas, in contrast to similar available satellite products restricted over land surfaces (Ginoux et al., 2012), thus making feasible a thorough description of dust loads not only over the sources but also over downwind regions where transport, from short- to long-range, is taking place. Reanalysis datasets, spanning through decades and available at high temporal frequency, can fulfill such tasks; however, their coarser spatial resolution imposes a restriction when investigating mineral loads' features at finer spatial scales. Our developed DOD product aims at complementing existing observational gaps and can be exploited in a variety of studies related to dust climatology and trends, evaluation of atmospheric-dust models, estimation of dust radiative effects and assessment of the associated impacts as well as on improving dust forecasting and monitoring via data assimilation.

The core concept of the applied methodology relies on the utilization of MODIS AOD and MERRA-2 DOD-to-AOD ratio for the derivation of DOD on MODIS swaths. Nevertheless, two prior steps have been done ensuring the necessity and the validity of the proposed method. First, the MODIS AODs have been compared against the corresponding MERRA-2 outputs in order to justify that these two datasets are not similar, thus avoiding the consideration of a dataset of low temporal frequency (one overpass per day) rather than hourly reanalysis outputs. According to our results, the spatial AOD global patterns are commonly reproduced by MODIS and MERRA-2; however, there are evident differences in the aerosols loads. For example, MERRA-2 shows a higher AOD across N. Africa by up to 0.25, and over other dusty regions. Over biomass burning areas (e.g. Amazon, Central Africa) and industrialized regions (E. Asia) the AOD can be smaller in MERRA-2 by up to 0.3. Over oceans, the majority of MERRA-2 – MODIS differences are very small, except in the case of the Tropical Atlantic Ocean where the MODIS AODs are higher than the corresponding simulated levels by up to 0.05.

The second prior step comprises the evaluation of the MERRA-2 MDF against reference values obtained by the columnar integration of quality assured dust and non-dust CALIOP profiles. Over dust-abundant areas extending across the "dust belt", MERRA-2 reproduces adequately the magnitude of dust portion as indicated by the calculated primary statistics (bias, FB, FGE) with the maximum underestimations (up to 10%) being observed in Asian deserts. The agreement between MERRA-2 and CALIOP is reduced in the main dust regions of N. America and in the S. Hemisphere. Regarding the temporal covariation of the observed and simulated dust portions, over the period 2007-



2015, moderate R values (up to 0.5) are computed above the sources, attributed to the high
spatiotemporal variability of the emission processes. On the contrary, the correlation increases
substantially (up to 0.9) over downwind maritime regions (Tropical Atlantic Ocean, north Pacific
Ocean, Arabian Sea, Mediterranean Sea) where the main dust transport paths are recorded. Apart
from the geographical dependency of the level of agreement between MERRA-2 and CALIOP DOD-
to-AOD ratios, we also investigated the impact of the spatial representativeness of the CALIOP
observations. Through this analysis, we revealed that for an increasing number of CALIOP L2
profiles (from 1 to 23), aggregated for the derivation of the 1° x 1° grid cell, the computed metrics
converge towards their ideal scores.
Finally, the obtained MIDAS DOD was evaluated against AERONET retrievals and compared
with CALIOP and MERRA-2 DODs. AEROENT observations were processed to minimize the
contribution of other aerosol species and assuming that dust loads are mainly consist of coarse
particles (their radii is larger than the defined inflection point). Overall, the agreement between ~7300
MIDAS-AERONET pairs resides is very high (R=0.882), whereas the satellite DODs are lower by
~5% with respect to the ground-based ones. At station level, the R values are mainly above 0.8 at
most sites of the N. Hemisphere (except western US) while they don't exceed 0.5 in the S.
Hemisphere. Moreover, positive MIDAS-AERONET deviations (up to 0.133) are mainly
encountered in N. Africa and Middle East in contrast to negative values (down to -0.154) recorded at
the remaining sites. Based on the annual and seasonal global DOD patterns averaged over the period
2007-2015 the locations with the maximum DODs are in a good agreement among the three datasets.
Nevertheless, in many regions (e.g., Bodélé, sub-Sahel, north Pacific Ocean) there are deviations on
the intensity of dust loads, attributed to the inherent weaknesses of DOD derivation techniques based
on different approaches. Despite the regional dependency of biases among the three datasets, the
collocated global long-term averaged DODs are very similar (0.029 for CALIOP, 0.031 for MERRA-
2 and MIDAS) and very close to those reported (0.030) by Ridley et al. (2016). In the S. Hemisphere
the corresponding levels (0.007-0.008) slightly differ for the three datasets, whereas in the N.
Hemisphere, CALIOP DODs (0.050) are lower by 10% with respect to MIDAS and MERRA-2
hemispherical averages (0.055).
As a demonstration of the MIDAS dataset, a brief discussion about dust loads' regime at global
scale is made by analyzing the annual and seasonal DOD patterns. The most pronounced dust activity
recorded in the Bodélé Depression of the northern Lake Chad Basin (DODs up to ~1.2), across the
Sahel (DODs up to 0.8), in western parts of the Sahara Desert (DODs up to 0.6), in the eastern parts
of the Arabian Peninsula (DODs up to ~1), along the Indus river basin (DODs up to 0.8) and in the
Taklamakan Desert (DODs up to ~1). On the contrary, the weaker emission mechanisms triggering
dust mobilization over the spatially limited sources of Patagonia, South Africa and interior arid areas



of Australia do not favor the accumulation of mineral particles at large amounts (DODs up to 0.4 at
local hotspots), even during high-dust seasons. Over oceans, the main pathways of long-range dust
transport are observed along the tropical Atlantic and the northern Pacific, revealing a remarkable
variation, within the course of the year, in terms of intensity, latitudinal position and range. Finally,
the Mediterranean and the Arabian Sea are affected by advected dust plumes originating from N.
Africa and Middle East, respectively. Based on the performed uncertainty analysis, the MIDAS DOD
product, within the course of the year, is highly reliable (less than 0.1 and 20% in absolute and relative
terms, respectively) over dust rich regions and becomes more uncertain (>60%) in areas where the
existence of dust loads is not frequent. This contradiction is interpreted by the stronger "signal" of
dust loads and the larger data availability thus converging towards lower measurement and sampling
uncertainties.
The exploitation of the MIDAS DOD product will be expanded in scheduled and under
preparation studies. At present, focus is given on: (i) the DOD climatology over dust sources and
downwind regions, (ii) the implementation of the MIDAS dataset in the DA scheme of the NMMB-
MONARCH model (Di Tomaso et al., 2017) and (iii) the estimation of dust radiative effects and the
associated impacts on solar energy production, in North Africa and Middle East, upgrading the work
of Kosmopoulos et al. (2018).

**Acknowledgments**
The DUST-GLASS project has received funding from the European Union's Horizon 2020 Research
and Innovation programme under the Marie Skłodowska-Curie grant agreement No. 749461. The
authors acknowledge support from the EU COST Action CA16202 "International Network to
Encourage the Use of Monitoring and Forecasting Dust Products (inDust). We would like to thank
the principal investigators maintaining the AERONET sites used in the present work. We thank the
NASA CALIPSO team and NASA/LaRC/ASDC for making available the CALIPSO products which
are used to build the LIVAS products and ESA which funded the LIVAS project (contract no.
4000104106/11/NL/FF/fk). We are grateful to the AERIS/ICARE Data and Services Center for
providing access to the CALIPSO data used and their computational center (http://www.icare.univ-
lille1.fr/, last access: 8 August 2019). Vassilis Amiridis acknowledges support by the European
Research Council (grant no. 725698, D-TECT). Eleni Marinou acknowledges support by the
Deutscher Akademischer Austauschdienst (grant no. 57370121). Carlos Pérez García-Pando
acknowledges support by the European Research Council (grant no. 773051, FRAGMENT), the AXA
Research Fund, and the Spanish Ministry of Science, Innovation and Universities (RYC-2015-18690
and CGL2017-88911-R) and the DustClim Project as part of ERA4CS, an ERA-NET initiated by JPI
Climate, and funded by FORMAS (SE), DLR (DE), BMWFW (AT), IFD (DK), MINECO (ES), ANR



(FR) with co-funding by the European Union (Grant 690462). We acknowledge support of this work
by the project "PANhellenic infrastructure for Atmospheric Composition and climatE change" (MIS
5021516) which is implemented under the Action "Reinforcement of the Research and Innovation
Infrastructure", funded by the Operational Programme "Competitiveness, Entrepreneurship and
Innovation" (NSRF 2014-2020) and co-financed by Greece and the European Union (European
Regional Development Fund). The authors would like to thank Dr. Andrew Mark Sayer for his
valuable and constructive comments. The authors would like also to thank Thanasis Georgiou for
developing the ftp server where the MIDAS dataset is stored.

**Data availability**

The MIDAS dataset is available at https://doi.org/10.5281/zenodo.3719222

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





**Table 1:** Planetary (GLB), hemispherical (NHE and SHE) and regional DOD averages, representative for the period
2007-2015, based on collocated CALIOP, MERRA-2 and MIDAS 1°x1° data. Within the brackets are given the minimum
and maximum limits. The regional averages have been calculated following the upper branch (first temporal averaging
and then spatial averaging) in Figure 5 of Levy et al. (2009). The full names of the acronyms for each sub-region are
given in the caption of Figure S7.

| REGION | CALIOP | MERRA-2 | MIDAS |
|--------|--------|---------|-------|
| GLB | 0.029 [0.027-0.033] | 0.031 [0.027-0.035] | 0.031 [0.028-0.035] |
| NHE | 0.050 [0.048-0.062] | 0.055 [0.049-0.065] | 0.055 [0.051-0.065] |
| SHE | 0.008 [0.007-0.008] | 0.007 [0.006-0.008] | 0.007 [0.006-0.008] |
| ETA | 0.105 [0.083-0.172] | 0.095 [0.077-0.141] | 0.108 [0.086-0.163] |
| WTA | 0.027 [0.022-0.034] | 0.019 [0.016-0.024] | 0.021 [0.018-0.028] |
| MED | 0.072 [0.062-0.092] | 0.089 [0.079-0.102] | 0.091 [0.082-0.107] |
| GOG | 0.166 [0.085-0.292] | 0.275 [0.076-0.434] | 0.323 [0.097-0.488] |
| WSA | 0.259 [0.233-0.332] | 0.337 [0.309-0.388] | 0.306 [0.275-0.393] |
| SSA | 0.291 [0.237-0.397] | 0.263 [0.158-0.356] | 0.253 [0.163-0.355] |
| BOD | 0.309 [0.217-0.366] | 0.519 [0.393-0.637] | 0.598 [0.416-0.883] |
| NME | 0.218 [0.104-0.257] | 0.243 [0.142-0.252] | 0.296 [0.144-0.350] |
| SME | 0.212 [0.171-0.253] | 0.203 [0.176-0.258] | 0.186 [0.156-0.237] |
| CAS | 0.078 [0.051-0.090] | 0.139 [0.128-0.202] | 0.137 [0.106-0.184] |
| THA | 0.172 [0.112-0.204] | 0.143 [0.113-0.156] | 0.137 [0.079-0.155] |
| TAK | 0.372 [0.284-0.448] | 0.262 [0.234-0.320] | 0.144 [0.102-0.285] |
| GOB | 0.121 [0.090-0.156] | 0.120 [0.107-0.147] | 0.154 [0.073-0.146] |
| EAS | 0.089 [0.055-0.131] | 0.065 [0.049-0.080] | 0.077 [0.060-0.094] |
| WNP | 0.015 [0.013-0.019] | 0.026 [0.021-0.029] | 0.027 [0.022-0.030] |
| ENP | 0.008 [0.005-0.011] | 0.018 [0.014-0.019] | 0.018 [0.015-0.022] |
| SUS | 0.021 [0.011-0.031] | 0.028 [0.016-0.040] | 0.027 [0.012-0.042] |






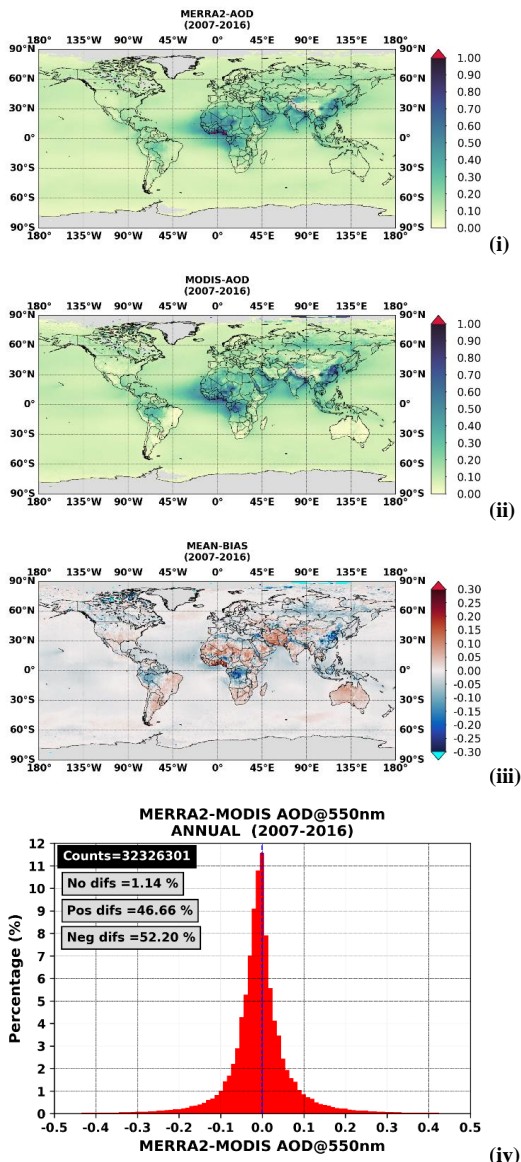

**Figure 1:** Annual geographical distributions of: **(i)** MERRA-2 AOD$_{550nm}$, **(ii)** MODIS-Aqua AOD$_{550nm}$ and **(iii)** MERRA2-MODIS AOD$_{550nm}$ biases, at 1° x 1° spatial resolution, averaged over the period 2007 – 2016. **(iv)** Relative frequency histogram of MERRA2-MODIS AOD$_{550nm}$ differences.

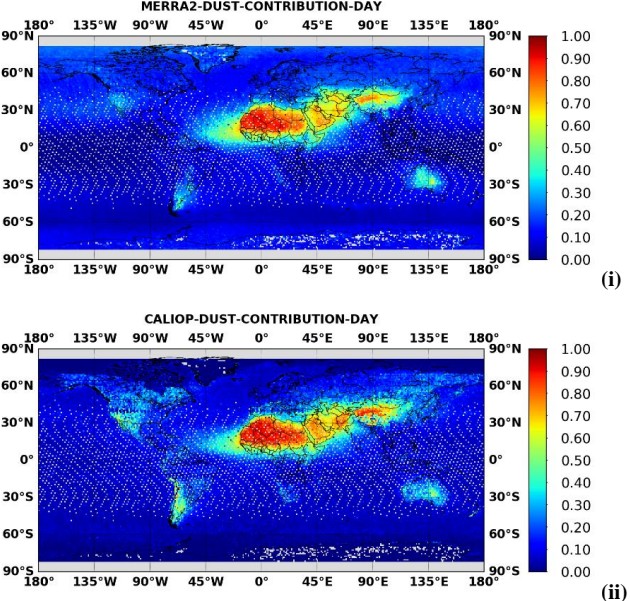

**Figure 2:** Annual geographical distributions of dust contribution to total aerosol optical depth, at 1° x 1° spatial resolution, based on: **(i)** MERRA-2 products at 550 nm and **(ii)** CALIOP retrievals at 532 nm, during daytime conditions, over the period 2007-2015.

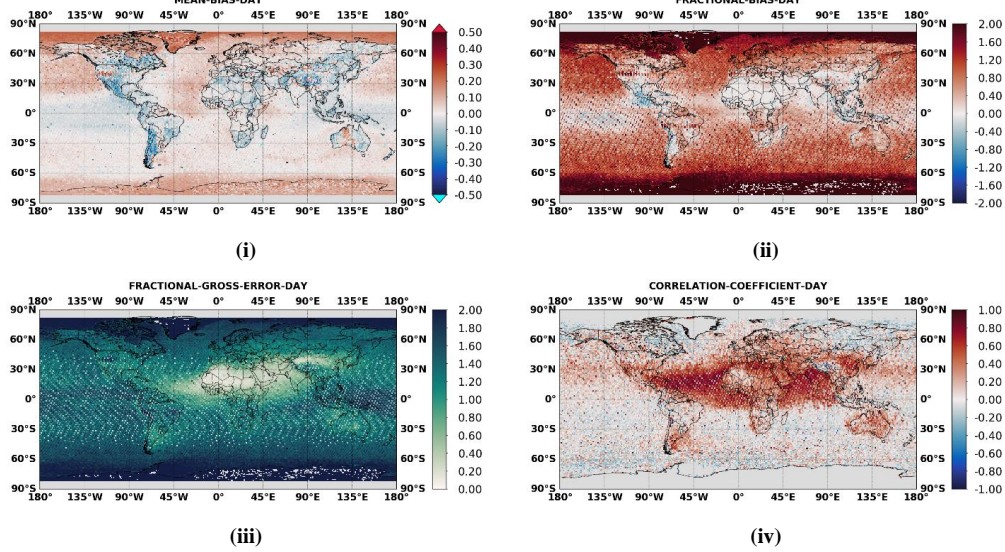

**Figure 3:** Annual geographical distributions illustrating the assessment of MERRA-2 dust-to-total AOD ratio versus CALIOP retrievals, during daytime conditions at 1° x 1° spatial resolution, according to the primary skill metrics of: **(i)** mean bias, **(ii)** fractional bias, **(iii)** fractional gross error and **(iv)** correlation coefficient, representative for the period 2007-2015.



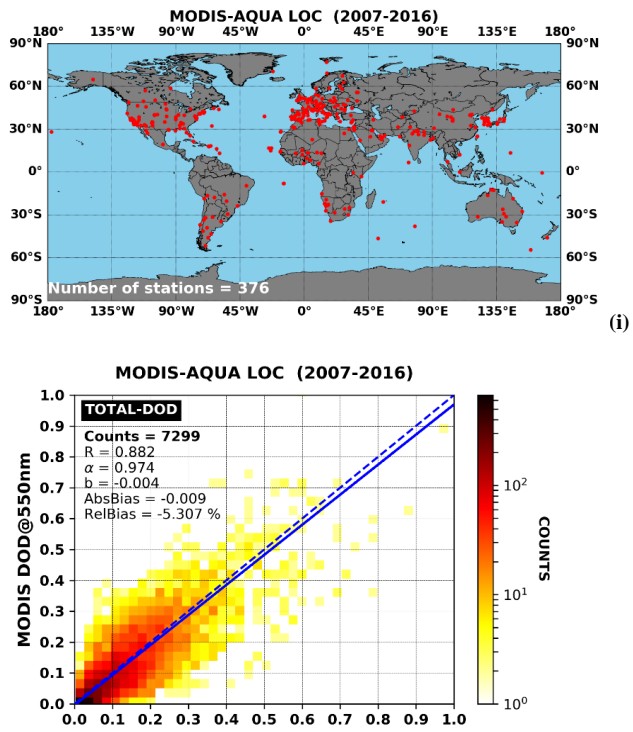

**Figure 4: (i)** AERONET sites where at least one pair of ground-based and spaceborne retrievals has been recorded, according to the defined collocation criteria, during the period 2007 – 2016. **(ii)** Density scatterplot between MODIS (y-axis) and AERONET (x-axis) dust optical depth at 550nm. The solid and dashed lines stand for the linear regression fit and equal line (y=x), respectively. LOC in the titles indicates that both land (L) and ocean (OC) MODIS retrievals are considered.



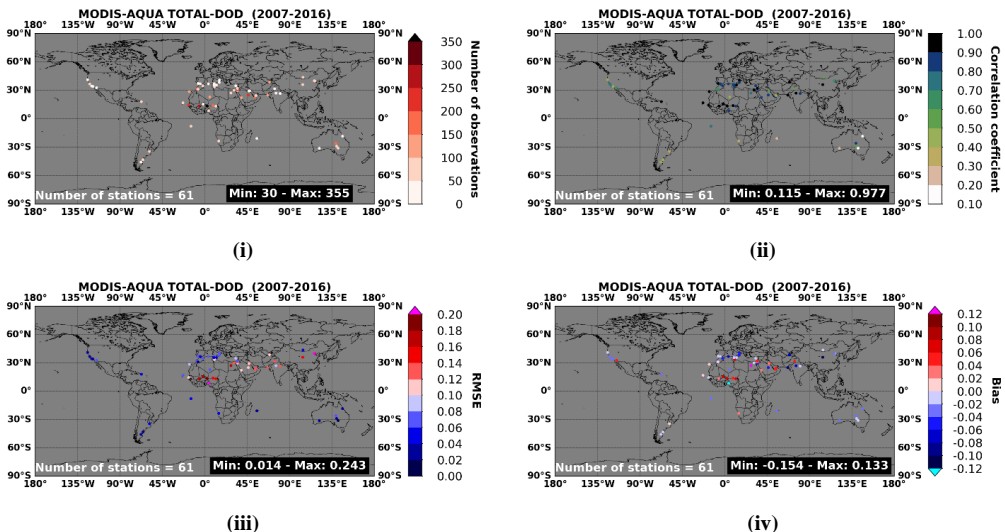

**Figure 5:** Scatterplot metrics between MODIS and AERONET DOD$_{550nm}$, at station level, during the period 2007 – 2016.
**(i)** Number of concurrent MODIS-AERONET observations, **(ii)** correlation coefficient, **(iii)** root mean square error and
**(iv)** bias defined as spaceborne minus ground-based retrievals. The obtained scores are presented for sites with at least 30
MODIS-AERONET matchups.



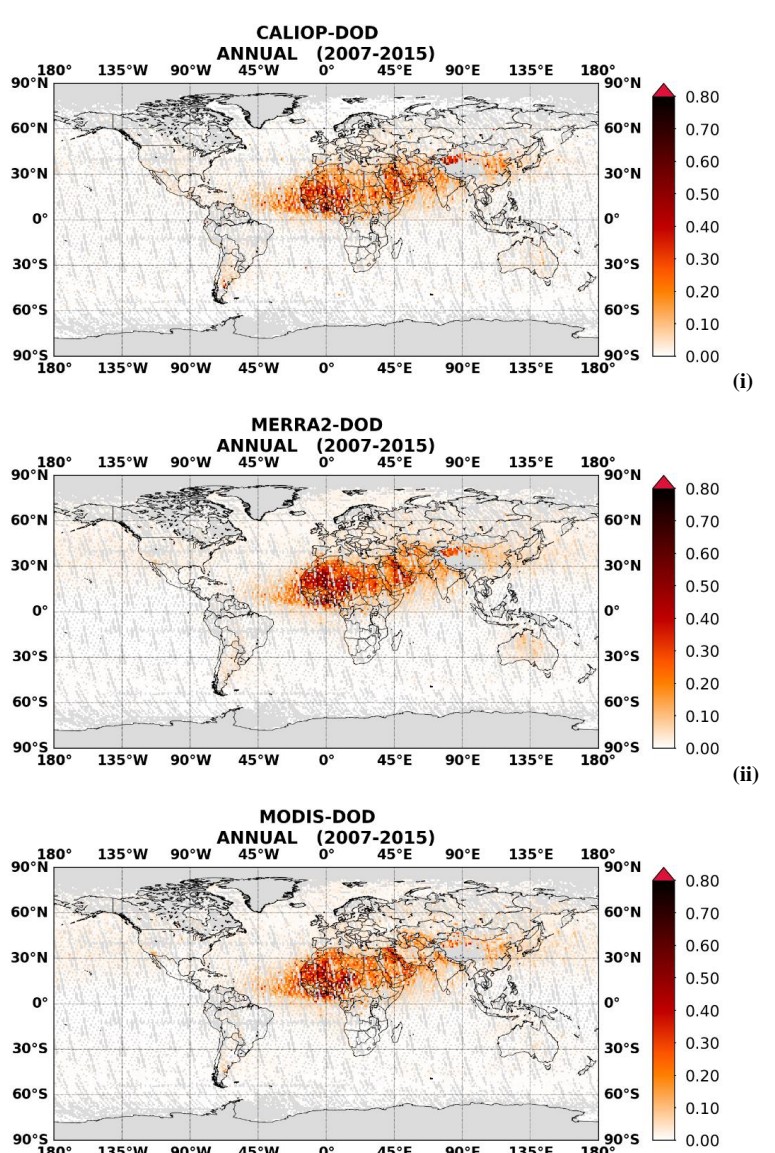

**Figure 6:** Long-term (2007 – 2015) average geographical distributions, at 1° x 1° spatial resolution, of daytime: **(i)** CALIOP $DOD_{532nm}$, **(ii)** MERRA-2 $DOD_{550nm}$ and **(iii)** MIDAS (MODIS) $DOD_{550nm}$.



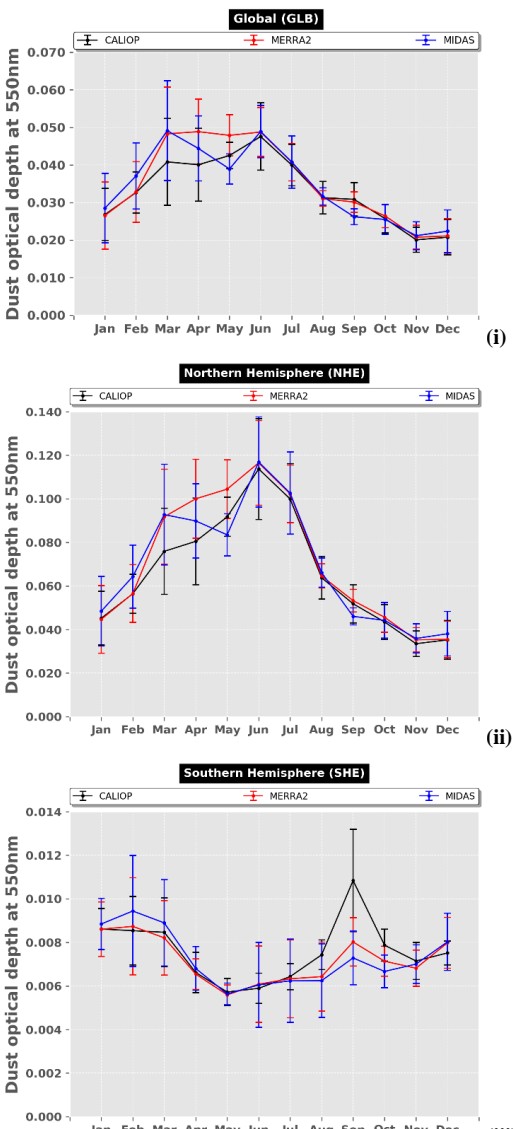

**Figure 7:** Intra-annual variability of CALIOP (black curve), MERRA-2 (red curve) and MODIS (blue curve) monthly DODs, regionally averaged over: **(i)** the whole globe (GLB), **(ii)** the Northern Hemisphere (NHE) and **(iii)** the Southern Hemisphere (SHE). The error bars correspond to the standard deviation computed from the interannual timeseries during the period 2007 – 2015.

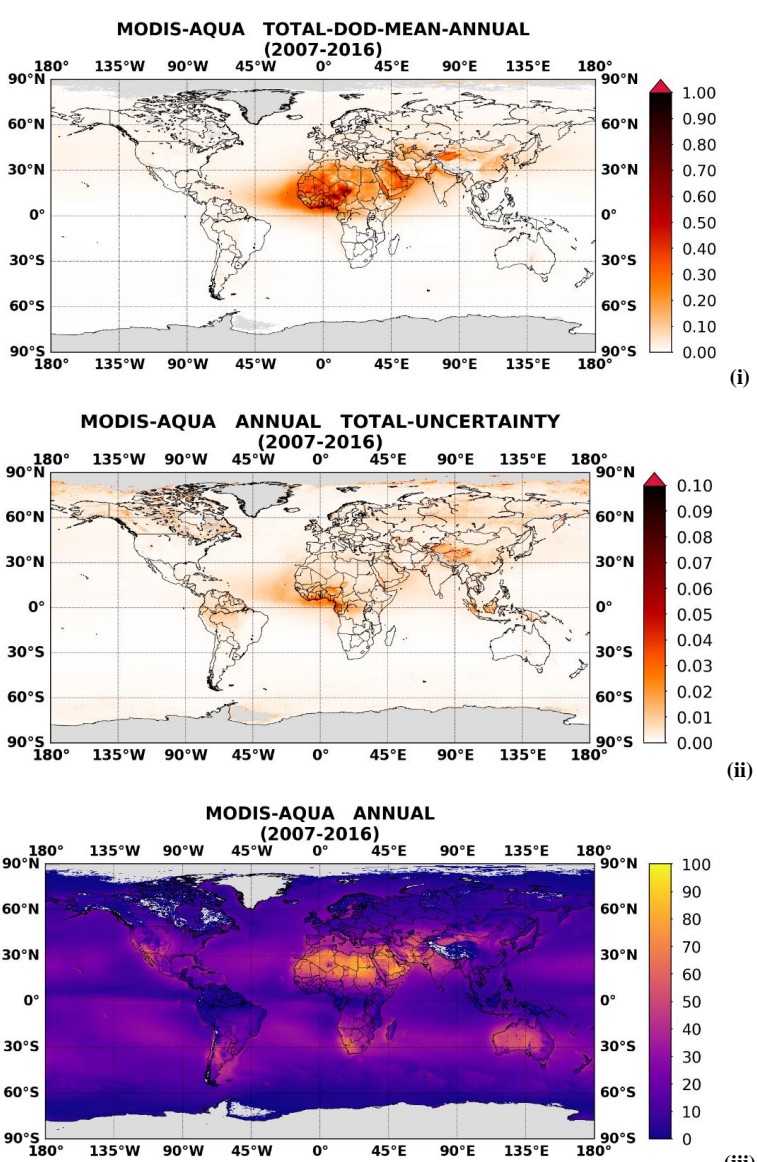

**Figure 8:** Annual geographical distributions, at 0.1° x 0.1° spatial resolution, of: **(i)** the climatological DODs, **(ii)** the absolute DOD uncertainty and **(iii)** the percentage availability of MODIS-Aqua retrievals with respect to the entire study period spanning from 1 January 2007 to 31 December 2016. Grey color represent areas with absence of data.

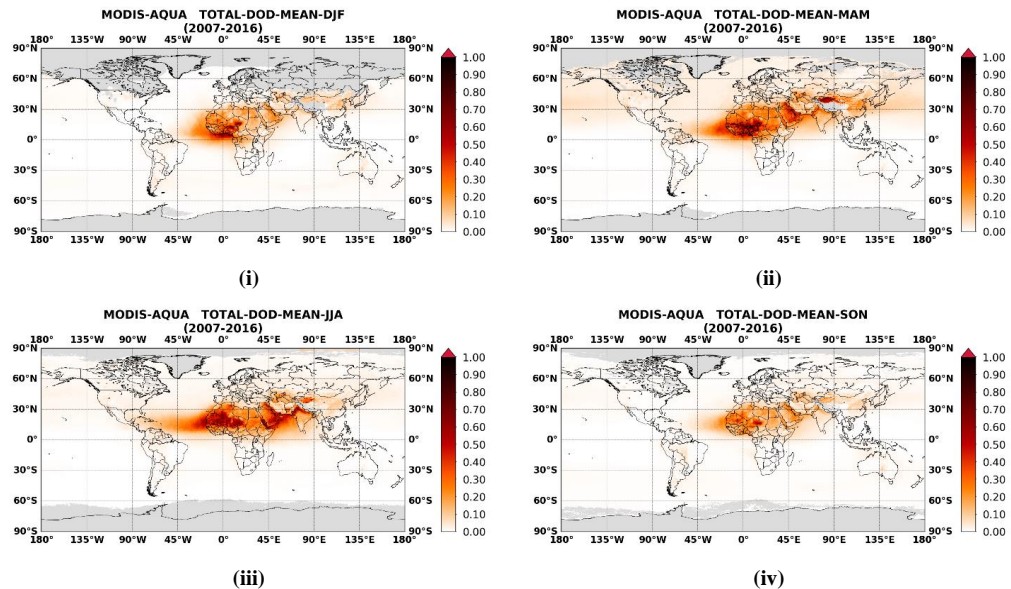

**Figure 9:** As in Figure 8-i but for: **(i)** December-January-February (DJF), **(ii)** March-April-May (MAM), **(iii)** June-July-August (JJA) and **(iv)** September-October-November (SON).