# Peer review of "ModIs Dust AeroSol (MIDAS): A global fine resolution dust optical depth dataset"

_Atmospheric Measurement Techniques, 2020_

## Referee Comment (RC1) · Anonymous Referee #1 · 6 Jul 2020

This study describes a new dust optical depth (DOD) data set, MIDAS, which is derived by taking MODIS (Aqua only) satellite-based aerosol optical depth (AOD) and the aerosol speciation from the MERRA2 reanalysis. CALIOP and AERONET are used for evaluation.

The manuscript is in scope for the journal, though would be a closer fit to the other Copernicus journal ESSD because it is mostly a data set description paper. The material is important because speciated AOD is one of the next frontiers for better climate and air quality applications of data sets. The quality of language and visuals is satisfactory overall, though some edits are needed, and figures 5, 7, 9 would benefit from labels being increased in font size (hard to read without zooming in). Some of the content in the Supplement should be in the main paper. Overall, I recommend major

revisions and would like to review the revision.

General comments:

The main technical weak point of this study is that all the observational data sets used are out of date: MODIS Collection 6 instead of 6.1; CALIOP version 3 instead of version 4; AERONET version 2 instead of version 3. So this affects the AOD source used (MODIS), the optical properties used for matching (AERONET), and the data sets used for evaluation (AERONET, CALIOP). Some of the differences between old and new versions are systematic. So it is not clear to me how different the derived data set, or the evaluation results, would be if the newest data versions were used. To my best knowledge all of these latest data versions have been available for 1.5 years or so (i.e. they are not that new), so it is unfortunate that outdated versions were used when this analysis was done. It sounds like the authors are using a post-processed CALIOP product from another group (LIVAS?) rather than the official NASA CALIOP data products, so maybe that can't be changed. But, if the authors intend for others to use MIDAS for scientific analyses, it would really be best to use the most up to date inputs. I know that this means more work downloading files and rerunning code but this should mostly be computer time if the analysis code has already been written. So my main recommendation is to do that. I guess it is up to the authors and editor to decide what is most reasonable here. The 2007-2016 time period could also possibly be extended, I see no reason why it couldn't cover more of the Aqua record. Longer time series are of course more beneficial for things like trend analyses.

Numbers are often given to too many significant digits. One example I'll mention again later includes referring to an offset as 4.264%. Including all these digits gives an unrealistic impression of the precision of these estimates: can you really say that the true population offset is 4.264% and not 4.265%? Is it important that it is 4.264% and not 4.265%? If the answer to either of these is no, this is an indication that there are too many significant digits being reported. The authors should consider all numbers presented in this manuscript. For this case, for example, I'd probably just say 4.3%. This

will also make the paper more readable.

I downloaded some MIDAS data from the link in the paper to have a look. The contents of those files seemed as described. I have four suggestions based on looking at these files:

1. I didn't see a MIDAS file version identifier, but the Readme file notes that some things are in testing or will be added in a future version. So it would be good to add a MIDAS version number somewhere in the filenames so the user can be sure which version of MIDAS they have (and which version of MIDAS technical documents such as this refer to). It may be unclear for the data user otherwise.

2. One issue with is that the files contain some negative AOD values, which are unphysical. This is a result of the Dark Target land AOD algorithm which allows small negative retrievals. However since this is unphysical I recommend that in the next version, the authors set these values to 0. This is one issue with the source data which is easily fixed.

3. It would also be useful to add an uncertainty estimate to each pixel. There is extensive discussion in the middle of the paper about uncertainty estimates, but these don't appear to have made it through to the data set itself, based on the files I looked at.

4. Finally, the files seem to contain some data fields inherited directly from the MODIS aerosol product, e.g. Angstrom exponents. As these are for total AOD and not dust AOD, I wonder if it would be better to remove these. Or, combine the Deep Blue land Angstrom exponent with one of the Dark Target ocean ones. There's also solar and sensor zenith angles, but I'm not sure what these are in there for. This would decrease the size of the archive to be downloaded somewhat.

My more specific comments are as follows:

Line 133: should the word "conclusions" be added before "are drawn"?

Lines 143-147: I suggest rewording this sentence. The Dark Target algorithms are really two different approaches as they have different bands used and completely different assumptions between them. Also, Deep Blue is over all snow-free land, not just bright deserts. So really it is one water algorithm (Dark Target ocean) and two land algorithms (Dark Target land, and Deep Blue). It is probably worth acknowledging that there are other MODIS aerosol algorithms too (e.g. MAIAC), they are just not included in those files.

Line 153: I think the authors mean either "increasing pixel size" or "decreasing pixel resolution" here. Not "increasing pixel resolution", which is the opposite.

Line 206 and 212-215: note that the MODIS aerosol product is not assimilated. Rather, it is a neural network retrieval based on MODIS radiances that is assimilated. Not a neural network bias correction based on the MODIS retrieval. So the MODIS information going into MERRA2 is not the same as is being used as the main AOD data set here.

Line 333: note that Levy reference is only for Dark Target over land. For discussion of Deep Blue Angstrom exponent over land, see the Sayer et al (2013) paper that is cited later in the manuscript.

Line 357: I am not sure it makes to take the quadrature sum of DT and DB uncertainties when they are merged. This means the overall uncertainty is worse than either DT or DB. If you think the uncertainties on these algorithms are independent, then you are effectively averaging two observations which means the uncertainty in the sum should be divided by sqrt(2). Since the retrieval is the average of two algorithms then the uncertainty should represent the uncertainty on that average.

Line 386-389: If the output is at 0.1 degrees, then there should only be 1 retrieval in each (as the MODIS product is 10x10 km at nadir), so I don't understand this part about decreasing uncertainties when you have multiple retrievals. Or is this about when there are overlapping retrievals at the edge of swath from consecutive orbits? If

so, that should be stated. If this is about averaging to a coarser space/time scale, then I don't think it makes sense to use the root n factor here because we know there is high spatial correlation in the errors because the errors are mostly not noise.

Section 4.1: I am not sure that it is useful to compare total MERRA2 and MODIS AOD in this way. Or at least, the framing of the purpose here is not right. If there is a systematic disagreement then that tells you that there might be an error in the derived MERRA2 dust fraction as well. Would it not be more meaningful for the present analysis to compare MERRA2 and MODIS dust AOD rather than total? Or to report summary results of the evaluation of MODIS AOD against AERONET (from DT/DB team studies)? As written, section 4.1 doesn't fit well with the rest of the paper.

Section 4.3: The authors here frame the differences as if MIDAS is in error. However, unlike the direct-Sun AERONET AOD data, the AERONET almucantar scan retrievals used here have non-negligible uncertainties (which are not necessarily random). So some of the discrepancies and biases might in fact come from uncertainties in the AERONET DOD estimates. This was not directly discussed beyond a mention that the AERONET DOD estimates made here neglect fine-mode dust, although I think with the AE filtering this is likely to be a negligible effect in most cases.

Lines 666, 667: here the authors say that MERRA2 has "biases" and "overestimates" compared to CALIOP. It would be better to refer to positive and negative "offsets" or "differences" instead, because "bias" and "oversestimate" imply a problem and that CALIOP is the truth. Really none of the data sets are the truth and we are only making comparisons and not diagnosing errors. So more neutral language like "offsets" should be used here (and throughout), and terms like "bias" and "overestimate" should be avoided unless it involves a comparison with something that can be considered a reference truth. I mentioned only these examples although there are others in this section and through the paper where these or similar terms are used (and there are places where the wording is ok as well).

[Figure]

Line 716: authors should check and clarify which of the data sets corresponds to which number here. For example the wording implies that 4.264% is more than 9.405% which is obviously backwards.

Sections 4.2 to 4.5 were honestly a little hard to read because it's a large amount of text which is basically describing several figures and providing references. This also comprises about 11 of 28 pages of body text in the paper. I wonder if this can be streamlined a bit. The authors write that there will be a follow up paper looking at this same material in more detail as well. So I wonder if here it is best to just show figures and highlight where the data sets do not agree well (and maybe try to figure out why), as these are areas to focus future study on. That type of approach (figure out where and why there are differences) would also make the paper fit better in AMT.

Table 1: This is a bit of a sea of numbers. It is difficult to easily pull out the main message here. What is the main message here? Or is this just for reference? Given it relies on regional acronyms, it would be better to present the map defining the regions in the main paper rather than in the supplement.

Figures 2,3: most of the world here is in the 0-0.2 range in Figure 2, which is very hard to distinguish visually because it is different tones of blue. Figure 3 solves this but as it's a separate figure, it is card to glance back and forward. Also, I am not sure how helpful it is to show an annual map here because dust seasonality is strong. So I suggest making seasonal maps instead of annual to replace these figures, this would give more insights. I am also not sure whether the maps of FB and FGE are needed for this panel. Maybe just replace this with an 8-panel figure: left column is seasonal MDF, right column is seasonal MDF minus CALIOP dust fraction (i.e. mean bias)? Then Figure 3 could be 4 panels showing seasonal correlation coefficients? I think they are the most crucial metrics to show here because they show the level of consistency in typical dust fractio and the variation captured by MERRA2, which are what inform the DOD uncertainty here. The other panels could maybe move to the Supplement if the authors think they are useful. I know there are a few seasonal maps in the Supplement

but think the maps discussed above should be in the main paper.

Figure 4 (and text discussion): I don't think the linear regression is appropriate here, so it should be removed. Since the uncertainty on DOD is proportional to total AOD, it is likely that the assumptions of regression are violated. Also I don't think a global regression is useful because it is likely there are regional differences in the errors, meaning that the global regression line is not informative. Same comments apply to Figure S5 in the Supplement.

Figure 5: the circles are all too small to see.

Figure 8 (iii): this is the mean of the DOD uncertainties, right? Or is it the uncertainty on the mean DOD? This needs to be stated more clearly.

Figure S4: this illustrates a problem I have with the validation methodology. A 4 hour averaging window is pretty huge! And the time variation of AERONET DOD in that window can be much bigger than the AERONET uncertainty. So some of the disagreement seen in Figure 4 is due to this time mismatch. For this example the range of DOD in this window is about 0.09, or 40% of the average. This makes it hard to assess the performance of MIDAS. This is something that shouldn't be buried in the Supplement; I didn't see the mention of a 4-hour window in the main paper (if it is there, it is not clear) so the reader may not realise how big it is. Probably a smaller window is needed, and some filter based on AERONET time homogeneity. I know this will decrease the data volume, maybe a lot, but with such a big time variation in DOD within the window it makes the AERONET comparison a lot less useful for MIDAS evaluation.

On a non-scientific note, I thought the use of MIDAS as an acronym was amusing and a good choice.

---

## Referee Comment (RC2) · Anonymous Referee #2 · 6 Aug 2020

This manuscript describes the methodology to obtain a dust aerosol optical depth data set from MODIS total AOD combined with the use of MERRA-2 to determine the dust fraction in the AOD. This opens nice perspectives, offering specificity to the aerosols retrievals and this with global coverage once per day, nice horizontal resolution and a long time series.

Although the concepts at the basis of this work, the goal and the obtained data set are scientifically very good, the manuscript itself needs major revision (and I am willing to review the revised version if the Editor finds it needed). The manuscript is very long and some parts are pretty difficult to read, being very descriptive with many numbers. Some parts do not bring a lot to the manuscript, while being quite long. Also, there is a lack of consistency in terms used to refer to the products (see below), which renders

the reading a bit difficult. Ideas to improve this can be found in the different comments.

I have some major general comments then addition specific major comments, then some minor comments (editing / suggestions).

Major general comments:

1) There is no mention of the thermal infrared (TIR) based DOD data (SEVIRI and IASI - for IASI data is available in the climate data store). These are very interesting as the TIR is only sensitive to dust, but gives DOD at TIR wavelength which needs to be converted to visible (step that includes some assumptions on particle size and properties, but also a bunch of assumptions are needed in this work). I am not saying the study should be redone with a full comparison with TIR dust data (although that would be pretty interesting, see also a further comment on the comparisons undertaken in this work), but that when trying to obtain pure dust AOD one should at least mention the TIR DOD. For example, after lines 110-113 it would be nice to have some sentences describing what MIDAS data brings in addition to the TIR-based DOD (for example IASI is also long-term, global twice per day instead of once, and 12km ground resolution at nadir). To be perfectly clear, this is not me being sceptical about the scientific interest of this work, but I think that some information on other methods to obtain DOD from satellites should be added to the manuscript.

2) Data from CALIOP and MODIS are used, but there is a confusion as to which data exactly. Indeed, the authors use for CALIOP either the "official" CALIOP product from NASA, or the LIVAS product that some of the authors have previously developed, but both are referred to as "CALIOP", making it pretty difficult to keep track of things. It is a little bit the same for the MODIS "official" AOD product and the MIDAS here developed product, it needs thinking to be sure which one is referred to in the manuscript. I recommend to use the product names everywhere in the manuscript, to avoid any confusion: wherever referring to the "non-official" product, please use consistently LIVAS and MIDAS, while keep the instrument name for the "official" products. This also

includes the plot titles, legends and caption.

3) For the MERRA-2 dust fraction, please always use the acronym defined (MDF) or at least the same words, avoid using dust "portion" or other terms, for consistency and clarity.

4) Why do you use only the MODIS data from Aqua (and not Terra)?

5) Why old versions are used both for CALIOP and AERONET while the new versions exist for some time now?

6) I think that there are too many descriptions of different data sets and of different comparisons, each time with a long description of the geographical features. This makes the paper a bit difficult to read.

7) Many numbers are given with too many digits. Please try to only provide significant digits.

Major specific comments:

1) Line 33: "ground-truth AERONET-derived DODs" -> There is no "truth", any measurement has uncertainties and biases. In particular here the DOD derived from AERONET is a complex product with a number of assumptions and no-one should see it as "the truth"

2) Section 2.3 on CALIOP: It is a bit unclear to me how the CALIOP subtypes are used in LIVAS. I have the feeling that LIVAS is a different retrieval, not using the CALIOP "official" features/retrievals and therefore I would recommend to only mention here what is really needed to understand LIVAS. Otherwise it is a bit confusing.

3) Section 3.1 on the methodology: Do you see a discontinuity in the MODIS DOD linked to changing of MERRA-2 grid cell?

4) Lines 330 to 332: "Our approach avoids on purpose the inclusion of additional optical properties providing information on aerosol size (alpha) available from MODIS and

absorptivity (Aerosol Index) from OMI that are characterized by inherent limitations". -> This is probably very unclear for the non-specialist reader and comes a bit out of the blue. If OMI is mentioned, the authors should at least explicit why using OMI would make sense when looking for dust aerosols and what are the potential drawbacks with it, other than data availability.

5) Lines 359-361: "Here, we are using the same equations replacing AERONET AODs with those given by MODIS. This relies on the fact (results not shown here) that their averages are almost unbiased." -> A bit unclear. Average of MODIS AODs unbiased wrt AERONET? If yes, this feels a bit short and at least a reference to the MODIS validation should be given and this should be discussed. A quick search (https://doi.org/10.1016/j.atmosenv.2018.12.004) showed me that indeed on average along many years and globally the MODIS mean bias wrt AERONET is particularly low. However, at regional scale this is not always true. In particular over the dustiest regions (N Africa and Middle East) it seems that there are more outliers in the comparisons linked to a more difficult AOD retrieval from MODIS. I am not saying that the MODIS AOD should not be used to estimate the uncertainty on the AOD, but that it should be discussed a bit more.

6) Equations 4 and 5: Those do confirm my thoughts in the previous comment. Levy et al (2013) write that (for DT land) "69.4% of MODIS AOD fall within expected uncertainty of $\pm(0.05 + 15 \%)$." This is, I guess, the origin of equation 5 here (and something similar can be found for equation 4). This means, to me, that DT land has a mean bias of 0.05 wrt AERONET, in contradiction with the sentence above saying that there is no bias.

7) Equation 8: I would appreciate a plot here (of the data that lead to this equation), both to show how good the fit is (does it really need such a complex polynomial curve?) and show some values. I computed them myself from the equation to have a feeling. [MDF uncertainty]: [0 0,2]; [0,1 0,14]; [0,2 0,15]; [0,3 0,18]; [0,4 0,22]; [0,5 0,25]; [0,6 0,26]; [0,7 0,24]; [0,8 0,18]; [0,9 0,1]; [1 -0,01 ]; Those uncertainties are not negligible (especially at low MDF), however they are not discussed at all.

8) Section 4.1 (and also in the conclusion): I do not at all understand this section. Why should the MODIS and MERRA-2 AODs be different? That would underline problems in one of the data sets (at least). And why do they need to be different for this analysis to work? I would say the opposite, that MODIS and MERRA-2 AODs should be similar enough to allow for this work to be relevant. Overall, I find this section 4.1 quite confusing, I am unsure what the authors are trying to show and how it fits in the rest of the paper. I would better see here a short summary of the MERRA-2 AOD validation (with references). And then a short discussion how this will impact the MIDAS data set. Also, this section contains discussions linked to the MDF (dust emission in GOCART for example), which should be moved to the next section.

9) Lines 455 to 457: Is the MIDAS DOD expected to be overestimated because the GOCART model overestimates dust emissions?

10) Figure 3: High resolution figures are needed. Here if I zoom in (to see details discussed) it becomes blurry

11) Lines 485 to 490: I don't understand. If there is a bias of about 10% but other metrics show the algorithm performs well, then why is there a bias? This should be explained also in the manuscript.

12) Lines 493-494: the correlation between MERRA-2 and CALIOP (LIVAS??) is less good over dust source regions due to the high variability. This is linked by the authors to a poor behaviour of the model in these cases. Can't we also imagine that CALIOP is not perfect there, as the very thin ground coverage makes it miss many events? This is discussed a bit further (lines 521 onwards), but I think it would be good to also mention around lines 493-494 that CALIOP (LIVAS?) is also not perfect.

13) Lines 533-534: the underestimation of CALIOP with respect to AERONET, is it the official product or LIVAS? Here, in this section, it is very confusing. I think most of the section refers to LIVAS but this specific sentence to the official product. If this is indeed the case, then I do not see how this information (and the discussion following) is

useful here in the paper, where LIVAS and MERRA-2 are compared. That discussion is already in section 2.3 in a different formulation.

14) Lines 558-560: Does this mean that overall, only 10 to 20 CALIOP measurements per grid cell were averaged along 9 years? If yes, this is very low and I don't think it can be considered representative.

15) General on section 4.2: this section is quite long, it contains the description of the differences and some discussion about the origin of those differences, but no discussion on the implications of underlined shortcomings on the MIDAS data set? In particular, the underestimation of MERRA-2 over dust sources should be discussed in terms of "how will it affect the MIDAS DOD".

16) Section 4.3: Why redo a MODIS validation against AERONET (not bringing anything new)? I think there are enough papers on that to just refer to one and remove this part, making the paper a bit shorter and less confusing.

17) Figure 5: please change the colour scale for the correlation coefficient as it is now very difficult to see

18) Section 4.4: I think that somehow this section should show what the new MIDAS product brings. The comparisons are currently done in a way that gives the impression it's just another product but not really improved or different from MERRIS-2 or the LIVAS climatology. This is linked to the fact that averages over long periods are analysed, so at the end we are just comparing (validating?) climatologies from different products. As MIDAS is not meant to be a climatology, I would not do this kind of comparisons a big point in the paper, but I would instead emphasize what MIDAS gives that those other products can't give. And validate the product at its resolution - but this is done in the comparison with AERONET.

19) Line 648-649: "the study period extends from 2007 to 2015, driven again from CALIOP's temporal availability" -> this is very confusing... CALIOP is still running...

so the authors probably mean LIVAS availability. This is only one of the many examples where it is not clear which data is referred to, leaving the reader in possible misunderstanding.

20) Figure 6: Why do we see orbit-like features on a 9 years average?

21) Line 673: "CALIOP underestimates AOD over the Sahara" -> again, I think this is the official CALIOP product, right? So how is it relevant here where comparing LIVAS? Same comment/question further, line 677: how does the CALIOP misclassification of clouds impact the LIVAS product? Overall in this section I have a feeling that there is discussion of both CALIOP and LIVAS but I can't see which is which and I am very confused as to what is really important for the work presented here.

22) Lines 686-688: I don't see the point of this sentence

23) Line 732: any explanation for the local minimum of MIDAS in May? This is very surprising.

24) Figure 8: Why is the uncertainty higher off the west coast of N Africa than inland?

Minor comments / suggestions:

1) Line 55: "Gobbi" -> Gobi

2) Reference list lines 87-88 -> This list is clearly not aiming at being exhaustive (which is understandable) but here about half the references (and the newest) are work from the (co-) authors of this paper, while overall I don't think they really do represent half the work on dust aerosols from space, and certainly not recently. Maybe it would be best to cite a review paper?

3) Line 100 correct reference is Di Tomaso... She is one of the co-authors...

4) Line 133: "Finally, the main findings are summarized and are drawn" -> I think this needs rephrasing

5) Line 142: "MODIS is mounted on the NASA's twin polar satellites Terra and Aqua acquiring high-quality aerosol data since 2000 and 2002, respectively, while thanks to its wide swath ($\sim$2330 km) provides near-global observations, almost on a daily basis" -> I think there's something wrong in the tenses

6) Line 207: "Over oceans, are also used AVHRR radiances" -> This reads weird, I suggest avoiding the passive formulation

7) Line 289: "requires the of SSA" -> I think a word is missing

8) Line 347: "in which $\sim$68% of the MODIS-AERONET AOD differences fall within" -> I think it needs rephrasing

9) Line 375: "higher or equal than" -> higher to or equal than?

10) Line 391: "These two uncertainty quantities" -> values?

11) Line 398: "On the following sections," -> In?

12) Line 405: "since a climatological study it is the scientific topic of the companion paper" -> reads weird. Rephrase? Or at least remove "it".

13) Lines 460: by "enormous number of pairs" do you mean that the histogram contains all the single comparisons and not just the time average comparison? This is unclear in the text.

14) Lines 471-472: "showing the ability of MERRA-2 in reproducing the integrated aerosol fields." -> This belongs to section 4.1 on AOD, not 4.2 on dust fraction

15) Line 480: the terms fractional bias (FB) and fractional gross error (FGE) should be a bit explained, those are not so standard statistics I think

16) Line 580: "discussed" -> described?

17) Line 582: "while for the derived DOD to check the validity of our approach" -> needs rephrasing

18) Line 583-584: "At first, a short discussion is made on the" -> I think "made" can't be used in that sense

19) Lines 592-593 "and the consideration of AERONET data." -> please rephrase

20) Line 600: "slight" -> slightly?

21) Line 600 (end) to 604: I think these sentences belong more to the methods section

22) Line 624: "fine DOD on AERONET" -> in?

23) Line 625-626:" but its contribution to the total dust AOD it is difficult and probably impossible to be quantified" -> remove "it"?

24) Lines 662-663: "it is apparent a very good agreement" -> A very good agreement is observed?

25) Line 671: "relied" -> relying?

26) Line 675: "works" -> work?

27) Line 679: "All these aspects, most likely met over dust sources" -> Please rephrase

28) Line 682-683: "Across the Sahel, CALIOP provides higher DODs (mainly up to 0.2) both against simulated and satellite products" -> confusing, CALIOP is a satellite, so maybe use here MODIS or MIDAS?

29) Line 792: "since it is not expected the accumulation of dust" -> since the accumulation of dust is not expected

30) Line 909: "AEROENT" (typo)

31) Line 910: "assuming that dust loads are mainly consist of" -> please rephrase

32) Line 912: remove "resides"?

---

## Author Comment (AC1) · 8 Oct 2020

We would like to thank the Reviewer for his/her thorough report that helped us improving the quality of our study. Through his/her constructive comments and suggestions the submitted manuscript has been updated significantly. Below are given point-by-point replies (regular font) to the comments (bold font) raised by the Reviewer.

**Reviewer #1**

**This study describes a new dust optical depth (DOD) data set, MIDAS, which is derived by taking MODIS (Aqua only) satellite-based aerosol optical depth (AOD) and the aerosol speciation from the MERRA2 reanalysis. CALIOP and AERONET are used for evaluation.**

**The manuscript is in scope for the journal, though would be a closer fit to the other Copernicus journal ESSD because it is mostly a data set description paper. The material is important because speciated AOD is one of the next frontiers for better climate and air quality applications of data sets. The quality of language and visuals is satisfactory overall, though some edits are needed, and figures 5, 7, 9 would benefit from labels being increased in font size (hard to read without zooming in). Some of the content in the Supplement should be in the main paper. Overall, I recommend major revisions and would like to review the revision.**

We agree with the Reviewer that our study fits well also with the scope of ESSD. Actually, the manuscript had been submitted to ESSD. The problem with ESSD was the long delay (3 months +) finding an editor. For this reason, we took the decision to withdraw the paper and resubmit it to AMT. Regarding the quality of the figures, we have reproduced all of them and their illustration has been improved.

**General comments:**

**The main technical weak point of this study is that all the observational data sets used are out of date: MODIS Collection 6 instead of 6.1; CALIOP version 3 instead of version 4; AERONET version 2 instead of version 3. So this affects the AOD source used (MODIS), the optical properties used for matching (AERONET), and the data sets used for evaluation (AERONET, CALIOP). Some of the differences between old and new versions are systematic. So it is not clear to me how different the derived data set, or the evaluation results, would be if the newest data versions were used. To my best knowledge all of these latest data versions have been available for 1.5 years or so (i.e. they are not that new), so it is unfortunate that outdated versions were used when this analysis was done. It sounds like the authors are using a post-processed CALIOP product from another group (LIVAS?) rather than the official NASA CALIOP data products, so maybe that can't be changed. But, if the authors intend for others to use MIDAS for scientific analyses, it would really be best to use the most up to date inputs. I know that this means more work downloading files and rerunning code but this should mostly be computer time if the analysis code has already been written. So my main recommendation is to do that. I guess it is up to the authors and editor to decide what is most reasonable here. The 2007-2016 time period could also possibly be extended, I see no reason why it couldn't cover more of the Aqua record. Longer time series are of course more beneficial for things like trend analyses.**

We decided to follow the reviewer's suggestion and in the revised manuscript we have used the MODIS-Aqua C061 data as well as the AERONET Version 3 retrievals. Moreover, the temporal availability of the MIDAS dataset has been extended from 10 (2007-2016) to 15 years (2003-2017). Therefore, the major comment raised by the Reviewer has been addressed adequately to our opinion. For the evaluation of the MDF we have used the CALIOP data which have been post-processed from our group and are provided via the LIVAS database (Amiridis et al., 2015). In the submitted manuscript, we stated (Lines $248 - 250$) the published works describing the methodology for the derivation of the pure dust product (accounting for dust plus its portion from dust mixtures; Amiridis et al., 2013) as well as the series of filters applied in order to analyze only the quality assured CALIOP profiles (Marinou et al., 2017). The aforementioned techniques are also briefly discussed in our manuscript (Section 2.3). The in-house developed LIVAS database has been built using

CALIOP V3 data and its temporal availability spans from 2007 to 2015. Currently, the group responsible for the ESA-LIVAS database is working on the development of an updated version, covering the entire CALIPSO CALIOP observational period, in which the CALIOP V4.2 profiles are used. We acknowledge that there is a confusion to the reader regarding the terms "CALIOP" and "LIVAS" which has been addressed in the revised document following the recommendation made also by the Reviewer 2.

**Numbers are often given to too many significant digits. One example I'll mention again later includes referring to an offset as 4.264%. Including all these digits gives an unrealistic impression of the precision of these estimates: can you really say that the true population offset is 4.264% and not 4.265%? Is it important that it is 4.264% and not 4.265%? If the answer to either of these is no, this is an indication that there are too many significant digits being reported. The authors should consider all numbers presented in this manuscript. For this case, for example, I'd probably just say 4.3%. This will also make the paper more readable.**

We agree with Reviewer. We have kept only one digit in all numbers mentioned in the text.

**I downloaded some MIDAS data from the link in the paper to have a look. The contents of those files seemed as described. I have four suggestions based on looking at these files:**

1. **I didn't see a MIDAS file version identifier, but the Readme file notes that some things are in testing or will be added in a future version. So it would be good to add a MIDAS version number somewhere in the filenames so the user can be sure which version of MIDAS they have (and which version of MIDAS technical documents such as this refer to). It may be unclear for the data user otherwise.**

We agree with the Reviewer that it was an omission from our side not including a file version identifier. We have changed the filenames by adding the MIDAS Version (V1) while the necessary notification is given in the new README file.

2. **One issue with is that the files contain some negative AOD values, which are unphysical. This is a result of the Dark Target land AOD algorithm which allows small negative retrievals. However since this is unphysical I recommend that in the next version, the authors set these values to 0. This is one issue with the source data which is easily fixed.**

We prefer to keep the negative values and give the option to the user to decide how he/she will treat them. For example, the inclusion of negative AOD values (reducing the positive biases in low-AOD conditions according to previous evaluation studies) in the calculation of long-term averages will give more "accurate" results. On the other hand, for the calculation of temporal or spatial, median or geometrical mean values the negative AODs can be replaced with very small positive values as it has been done in Sayer and Knobelspiesse (2019).

3. **It would also be useful to add an uncertainty estimate to each pixel. There is extensive discussion in the middle of the paper about uncertainty estimates, but these don't appear to have made it through to the data set itself, based on the files I looked at.**

The pixel-level DOD uncertainty has been added in the netcdf files.

4. **Finally, the files seem to contain some data fields inherited directly from the MODIS aerosol product, e.g. Angstrom exponents. As these are for total AOD and not dust AOD, I wonder if it would be better to remove these. Or, combine the Deep Blue land Angstrom exponent with one of the Dark Target ocean ones. There's also solar and sensor zenith angles, but I'm not sure what these are in there for. This would decrease the size of the archive to be downloaded somewhat.**

As correctly stated by the Reviewer, the Ångström exponents are related to AOD and not to DOD. We are storing them in the MIDAS netcdf files in case where a user would like to work only with AODs (also available in the netcdf files) and use these size parameters in parallel for a discrimination between coarse- or fine-particles dominant conditions. We think that it is better not to merge ocean and land Ångström exponents because they are provided at different wavelength pairs and this might confuse the users. The solar and sensor zenith angles are required for the estimation of the air mass factor (AMF, Eq. 6) according to Sayer et al. (2013) as clearly stated in the manuscript. Each MIDAS daily file has a size of ~10MB which is not "prohibitive" for a fast downloading.

**My more specific comments are as follows:**

**Line 133: should the word "conclusions" be added before "are drawn"?**

The missing word has been added.

**Lines 143-147: I suggest rewording this sentence. The Dark Target algorithms are really two different approaches as they have different bands used and completely different assumptions between them. Also, Deep Blue is over all snow-free land, not just bright deserts. So really it is one water algorithm (Dark Target ocean) and two land algorithms (Dark Target land, and Deep Blue). It is probably worth acknowledging that there are other MODIS aerosol algorithms too (e.g. MAIAC), they are just not included in those files.**

The sentence mentioned by the Reviewer has been rewritten in the revised manuscript as follows:

"*The derivation of AOD is achieved through the implementation of two retrieval algorithms based on the Dark Target (DT) approach, valid over oceans (Remer et al., 2002; 2005; 2008) and vegetated continental areas (Levy et al., 2007a; 2007b; 2010) but relying on different assumptions and bands, or the Deep Blue (DB) approach (Hsu et al., 2004; Sayer et al., 2013) over arid and semi-arid surfaces.*"

We don't see the point of mentioning other MODIS aerosol algorithms in Section 2.1 since it is discussed only the standard product which has been processed in our analysis.

**Line 153: I think the authors mean either "increasing pixel size" or "decreasing pixel resolution" here. Not "increasing pixel resolution", which is the opposite.**

We have corrected the sentence as suggested.

"*Each swath is composed by 203 x 135 retrievals, of increasing pixel size from the nadir view (10 km x 10 km) towards the edge of the satellite scan (48 km x 20 km), in which a Quality Assurance (QA) flag is assigned (Hubanks, 2018).*"

**Line 206 and 212-215: note that the MODIS aerosol product is not assimilated. Rather, it is a neural network retrieval based on MODIS radiances that is assimilated. Not a neural network bias correction based on the MODIS retrieval. So the MODIS information going into MERRA2 is not the same as is being used as the main AOD data set here.**

We have modified the relevant part of the text. Below is given the paragraph in the revised document.

"*For aerosol data assimilation, the core of the utilized satellite data is coming from the MODIS instrument multichannel radiances in addition to observational geometry parameters, cloud fraction and ancillary wind data. Over oceans, AVHRR radiances are used as well, from January 1980 to August 2002, and over bright surfaces (albedo > 0.15) the non-bias-corrected AOD (February 2000 – June 2014) retrieved for the Multiangle Imaging SpectroRadiometer (MISR; Kahn et al., 2005) is assimilated. Apart from spaceborne radiances and retrievals, the Level 2 (L2) quality-assured AERONET retrievals (1999 – October 2014; Holben*

*et al., 1998) are integrated in the MERRA-2 assimilation system (Goddard Aerosol Assimilation System, GAAS) which is presented in Randles et al. (2017; Section 3). The cloud-free MODIS (above dark target continental and maritime areas, Collection 5) and AVHRR (above oceanic regions) radiances are used for the derivation of bias-corrected AODs, via a neural net retrieval (NNR), adjusted to the log-transformed AERONET AODs.*"

**Line 333: note that Levy reference is only for Dark Target over land. For discussion of Deep Blue Angstrom exponent over land, see the Sayer et al (2013) paper that is cited later in the manuscript.**

The sentence has been rephrased to:

"*Previous evaluation studies (Levy et al., 2013; Sayer et al., 2013) have shown that size parameters acquired by MODIS are highly uncertain, particularly over land and at low AOD conditions.*"

**Line 357: I am not sure it makes to take the quadrature sum of DT and DB uncertainties when they are merged. This means the overall uncertainty is worse than either DT or DB. If you think the uncertainties on these algorithms are independent, then you are effectively averaging two observations which means the uncertainty in the sum should be divided by sqrt(2). Since the retrieval is the average of two algorithms then the uncertainty should represent the uncertainty on that average.**

The merged AOD uncertainty in the revised document is calculated based on the following formula:

$$\Delta(AOD_{DTDB-Land}) = \pm \frac{\sqrt{[\Delta(AOD_{DT-Land})]^2 + [\Delta(AOD_{DB-Land})]^2}}{2}$$

which is the uncertainty of the mean of DT and DB using the quadrature.

**Line 386-389: If the output is at 0.1 degrees, then there should only be 1 retrieval in each (as the MODIS product is 10x10 km at nadir), so I don't understand this part about decreasing uncertainties when you have multiple retrievals. Or is this about when there are overlapping retrievals at the edge of swath from consecutive orbits? If so, that should be stated. If this is about averaging to a coarser space/time scale, then I don't think it makes sense to use the root n factor here because we know there is high spatial correlation in the errors because the errors are mostly not noise.**

In this sentence we are describing the calculation of the DOD uncertainty at each pixel and at various temporal scales (i.e., monthly, seasonally, annually). Therefore, we are dealing with a time-series in which the applied algorithm (when possible) over land through time (i.e., from day-to-day) can switch from DT to DB depending on the NDVI threshold (see Sayer et al., 2014) while over oceans the AODs are retrieved always via the DT-Ocean algorithm.

**Section 4.1: I am not sure that it is useful to compare total MERRA2 and MODIS AOD in this way. Or at least, the framing of the purpose here is not right. If there is a systematic disagreement, then that tells you that there might be an error in the derived MERRA2 dust fraction as well. Would it not be more meaningful for the present analysis to compare MERRA2 and MODIS dust AOD rather than total? Or to report summary results of the evaluation of MODIS AOD against AERONET (from DT/DB team studies)? As written, section 4.1 doesn't fit well with the rest of the paper.**

We agree with the Reviewer and we have removed Section 4.1 from the revised manuscript. A similar comment regarding the usefulness of comparing MERRA-2 AOD versus MODIS in the current study has been raised also by the Reviewer 2. The intercomparison between MERRA-2 and MIDAS (MODIS) DODs, along with LIVAS (CALIOP), is already presented in Section 4.4.

**Section 4.3: The authors here frame the differences as if MIDAS is in error. However, unlike the direct-Sun AERONET AOD data, the AERONET almucantar scan retrievals used here have non-negligible uncertainties (which are not necessarily random). So some of the discrepancies and biases might in fact come from uncertainties in the AERONET DOD estimates. This was not directly discussed beyond a mention that the AERONET DOD estimates made here neglect fine-mode dust, although I think with the AE filtering this is likely to be a negligible effect in most cases.**

The comparison here includes two aspects. The use of the direct sun AERONET retrievals for AOD and the coincident inversions (using the SSA). Regarding AOD, the uncertainty is reported to be between 0.01 and 0.02, Eck et al. (1999). For the SSA, based on the results of Sinyuk et al. (2020) for the Mezaira site (i.e., predominance of dust aerosols), its uncertainty (being lower than 0.06) decreases significantly for increasing AODs. Since the $SSA_{675} - SSA_{440}$ difference (i.e. positive values) is used as a criterion for the discrimination of dust from sea-salt particles, the obtained SSA uncertainties, particularly those at 440nm, can affect the spectral signature of SSA and subsequently dust identification. Therefore, in some cases the AERONET DODs can be misclassified.

So to summarize, still AERONET AODs have lower uncertainty than the MODIS retrievals. For the case of the use of AERONET inversions, spectral SSA related uncertainties can lead to a misclassification of such cases.

**Lines 666, 667: here the authors say that MERRA2 has "biases" and "overestimates" compared to CALIOP. It would be better to refer to positive and negative "offsets" or "differences" instead, because "bias" and "overestimate" imply a problem and that CALIOP is the truth. Really none of the data sets are the truth and we are only making comparisons and not diagnosing errors. So more neutral language like "offsets" should be used here (and throughout), and terms like "bias" and "overestimate" should be avoided unless it involves a comparison with something that can be considered a reference truth. I mentioned only these examples although there are others in this section and through the paper where these or similar terms are used (and there are places where the wording is ok as well).**

We agree with the Reviewer's comment and we have made the appropriate modifications throughout the paper.

**Line 716: authors should check and clarify which of the data sets corresponds to which number here. For example the wording implies that 4.264% is more than 9.405% which is obviously backwards.**

Thanks for the correction!

**Sections 4.2 to 4.5 were honestly a little hard to read because it's a large amount of text which is basically describing several figures and providing references. This also comprises about 11 of 28 pages of body text in the paper. I wonder if this can be streamlined a bit. The authors write that there will be a follow up paper looking at this same material in more detail as well. So I wonder if here it is best to just show figures and highlight where the data sets do not agree well (and maybe try to figure out why), as these are areas to focus future study on. That type of approach (figure out where and why there are differences) would also make the paper fit better in AMT.**

We have made an effort to reduce the length of the text. As it concerns the interpretation of our findings we believe that we are providing all the necessary explanations without just describing plots. For example, there are statements about issues that can affect CALIOP (e.g., lidar ratio, total attenuation of the laser beam, cloud screening) and MODIS (e.g. surface reflectance) performance as well as MERRA-2 reliability (e.g., consideration only natural dust sources).

**Table 1: This is a bit of a sea of numbers. It is difficult to easily pull out the main message here. What is the main message here? Or is this just for reference? Given it relies on regional acronyms, it would be better to present the map defining the regions in the main paper rather than in the supplement.**

We are providing the long-term annual averages as well as their margins during the study period in order to have an overall view among the three DOD products at planetary scale, for the northern and southern hemisphere as well as for each sub-region. Figure S7 has been moved to the main text in order to help the reader. We think that it is useful for a scientist wanting to use such an information for a particular area or globally to have an easy and direct way of using these numbers.

**Figures 2,3: most of the world here is in the 0-0.2 range in Figure 2, which is very hard to distinguish visually because it is different tones of blue. Figure 3 solves this but as it's a separate figure, it is card to glance back and forward. Also, I am not sure how helpful it is to show an annual map here because dust seasonality is strong. So I suggest making seasonal maps instead of annual to replace these figures, this would give more insights. I am also not sure whether the maps of FB and FGE are needed for this panel. Maybe just replace this with an 8-panel figure: left column is seasonal MDF, right column is seasonal MDF minus CALIOP dust fraction (i.e. mean bias)? Then Figure 3 could be 4 panels showing seasonal correlation coefficients? I think they are the most crucial metrics to show here because they show the level of consistency in typical dust fraction and the variation captured by MERRA2, which are what inform the DOD uncertainty here. The other panels could maybe move to the Supplement if the authors think they are useful. I know there are a few seasonal maps in the Supplement but think the maps discussed above should be in the main paper.**

Following the suggestion made by the Reviewer, we have added in the Supplement a panel of figures presenting the biases and the correlation coefficients obtained on a seasonal basis. Also, the most important findings are briefly discussed in the main text. The FB and FGE metrics are less affected by outliers with respect to bias and serve as a complementary diagnostic tool.

**Figure 4 (and text discussion): I don't think the linear regression is appropriate here, so it should be removed. Since the uncertainty on DOD is proportional to total AOD, it is likely that the assumptions of regression are violated. Also I don't think a global regression is useful because it is likely there are regional differences in the errors, meaning that the global regression line is not informative. Same comments apply to Figure S5 in the Supplement.**

The way we define the uncertainty here is exactly the one that MODIS is using for AOD. It is a common practice to compare ground-based and spaceborne AODs (DODs in our case) through scatterplots. Figure 4 shows the overall comparison of MIDAS and AERONET DODs at global scale. Of course there are regional differences which are presented in the calculated metrics at station level in Figure 5. Therefore, all the necessary information is included. Figure S5 has been removed from the revised supplementary material.

**Figure 5: the circles are all too small to see.**

We have increased the size of the circles.

**Figure 8 (iii): this is the mean of the DOD uncertainties, right? Or is it the uncertainty on the mean DOD? This needs to be stated more clearly.**

We acknowledge that the description in the submitted document was not clear to the reader and for this reason we have modified accordingly the revised text. Figure 8-iii in the submitted document shows the uncertainty of the DOD average while in the revised text depicts the mean of the DOD uncertainties over the study period. Below is given the relevant part of the revised text.

"*Depending on the selected MODIS algorithm, the appropriate combination between AOD (Eqs. 4, 5, 6 and 7) and MDF (Eq. 8) uncertainties is applied to calculate the Δ(DOD) (Eq. 3) on each measurement (i.e., DOD) and at each grid cell. These pixel-level DOD uncertainties are averaged over the entire study period as well as for each season and the obtained findings will be discussed along with the global spatial patterns (Section 4.5) of dust optical depth in order to provide a measure of the reliability of the derived MIDAS DOD product.*"

**Figure S4: this illustrates a problem I have with the validation methodology. A 4-hour averaging window is pretty huge! And the time variation of AERONET DOD in that window can be much bigger than the AERONET uncertainty. So some of the disagreement seen in Figure 4 is due to this time mismatch. For this example, the range of DOD in this window is about 0.09, or 40% of the average. This makes it hard to assess the performance of MIDAS. This is something that shouldn't be buried in the Supplement; I didn't see the mention of a 4-hour window in the main paper (if it is there, it is not clear) so the reader may not realise how big it is. Probably a smaller window is needed, and some filter based on AERONET time homogeneity. I know this will decrease the data volume, maybe a lot, but with such a big time variation in DOD within the window it makes the AERONET comparison a lot less useful for MIDAS evaluation.**

It is true that the 4-hour time window is not the optimum and it would be better to be reduced down to ±30 minutes (a temporal margin applied in many evaluation studies). However, there are reasonable arguments which can support our approach. Please note that we are using the almucantar retrievals which have substantial less amount of data with respect to O'Neill retrievals or to sun-direct measurements. This volume of ground-based data is further suppressed when we are applying the criteria for the "determination" of AERONET DOD. By adding a time homogeneity criterion (which probably would be arbitrary), as suggested by the Reviewer, then more data are masked out from our sample. In our case, we had identified the MODIS-AERONET common pairs based on the ±30 min and ±1 hour time-window frames but the number of coincident observations, derived mainly at desert stations, was very small.

We believe that the Figure S4 should remain in the supplement rather than move it to the main text because it is just an illustration of the collocation method. Likewise, we would like to clarify that the MODIS map and the AERONET timeseries both refer to AOD and not DOD. The treatment of both datasets for the derivation of DOD is described sufficiently in the relevant sections of the paper.

**On a non-scientific note, I thought the use of MIDAS as an acronym was amusing and a good choice.**

Thank you!

---

## Author Comment (AC2) · 8 Oct 2020

**Reviewer #2**

We would like to thank the Reviewer for his/her constructive comments that helped us to improve the quality of our work as well as to clarify misleading points. Our replies (regular font) for each comment (bold font) are provided below.

This manuscript describes the methodology to obtain a dust aerosol optical depth data set from MODIS total AOD combined with the use of MERRA-2 to determine the dust fraction in the AOD. This opens nice perspectives, offering specificity to the aerosols retrievals and this with global coverage once per day, nice horizontal resolution and a long time series.

Although the concepts at the basis of this work, the goal and the obtained data set are scientifically very good, the manuscript itself needs major revision (and I am willing to review the revised version if the Editor finds it needed). The manuscript is very long and some parts are pretty difficult to read, being very descriptive with many numbers. Some parts do not bring a lot to the manuscript, while being quite long. Also, there is a lack of consistency in terms used to refer to the products (see below), which renders the reading a bit difficult. Ideas to improve this can be found in the different comments.

We would like to thank the Reviewer for his/her positive opinion about the scientific contribution and the importance of our study. Regarding the issues raised here, we would like to inform that we have made a major effort to reduce the length of the manuscript and shrink parts of the text in which many numbers are given. Moreover, a better clarification of the used/obtained datasets has been made thus addressing inconsistencies mentioned by the Reviewer. Our detailed replies are given in the relevant comments listed below.

I have some major general comments then addition specific major comments, then some minor comments (editing / suggestions).

**Major general comments:**

1. There is no mention of the thermal infrared (TIR) based DOD data (SEVIRI and IASI - for IASI data is available in the climate data store). These are very interesting as the TIR is only sensitive to dust, but gives DOD at TIR wavelength which needs to be converted to visible (step that includes some assumptions on particle size and properties, but also a bunch of assumptions are needed in this work). I am not saying the study should be redone with a full comparison with TIR dust data (although that would be pretty interesting, see also a further comment on the comparisons undertaken in this work), but that when trying to obtain pure dust AOD one should at least mention the TIR DOD. For example, after lines 110-113 it would be nice to have some sentences describing what MIDAS data brings in addition to the TIR-based DOD (for example IASI is also long-term, global twice per day instead of once, and 12km ground resolution at nadir). To be perfectly clear, this is not me being skeptical about the scientific interest of this work, but I think that some information on other methods to obtain DOD from satellites should be added to the manuscript.

We have added in the revised manuscript the missing information about DOD retrievals operating at TIR wavelengths, as correctly pointed out by the Reviewer. We find very interesting the idea of comparing MIDAS DOD against those provided by IASI and SEVIRI. Actually, it is a very nice perspective for a further exploitation of the MIDAS dataset!

2. Data from CALIOP and MODIS are used, but there is a confusion as to which data exactly. Indeed, the authors use for CALIOP either the "official" CALIOP product from NASA, or the LIVAS product that some of the authors have previously developed, but both are referred to as "CALIOP", making it pretty difficult to keep track of things. It is a little bit the same for the MODIS "official" AOD product and the MIDAS here developed product, it needs thinking to be sure which one is referred

to in the manuscript. I recommend to use the product names everywhere in the manuscript, to avoid any confusion: wherever referring to the "non-official" product, please use consistently LIVAS and MIDAS, while keep the instrument name for the "official" products. This also includes the plot titles, legends and caption.

All the necessary replacements, as suggested by the Reviewer, have been made throughout the revised manuscript (text, plots and captions).

**3. For the MERRA-2 dust fraction, please always use the acronym defined (MDF) or at least the same words, avoid using dust "portion" or other terms, for consistency and clarity.**

We think that it is pretty clear to the reader that MDF and MERRA-2 dust portion (or fraction) have the same meaning and there are not consistency or definition issues. However, in the revised manuscript the number of MDF "instances" has been increased at the expense of those of "dust portion (fraction)" trying at the same time to avoid the usage of this term very frequently which makes difficult, to our opinion, the readability of the text.

**4. Why do you use only the MODIS data from Aqua (and not Terra)?**

The obvious reason is that Aqua and CALIPSO are flying in the A-Train constellation which means that MODIS and CALIOP retrievals are almost coincident in temporal terms. This ensures that time departures between these two spaceborne sensors are not affecting our results in contrast to Terra which flies three hours earlier than Aqua. Nevertheless, the MODIS-Terra L2 data currently are processed and the derived MIDAS netcdf files will be uploaded as soon as possible in the same repository.

**5. Why old versions are used both for CALIOP and AERONET while the new versions exist for some time now?**

A same (similar) comment has been raised by the Reviewer 1. We are copying our reply below.

In the revised manuscript we have used the MODIS-Aqua C061 data as well as the AERONET Version 3 retrievals. Moreover, the temporal availability of the MIDAS dataset has been extended from 2007-2016 (10 years) to 2003-2017 (15 years). Therefore, the major comment raised by both Reviewers has been addressed adequately to our opinion. For the evaluation of the MDF we have used the CALIOP data which have been post-processed from our group and are provided via the LIVAS database (Amiridis et al., 2015). In the submitted manuscript, they are stated (Lines 248 – 250) the published works describing the methodology for the derivation of the pure dust product (accounting for dust plus its portion from dust mixtures; Amiridis et al., 2013) as well as the series of filters applied in order to analyze only the quality assured CALIOP profiles (Marinou et al., 2017). The aforementioned techniques are also briefly discussed in our manuscript (Section 2.3). The in-house developed LIVAS database has been built using CALIOP V3 data and its temporal availability spans from 2007 to 2015. Currently, we are working on the development of the updated LIVAS database, spanning from 2006 to 2020, in which the CALIOP V4 profiles are used.

**6. I think that there are too many descriptions of different data sets and of different comparisons, each time with a long description of the geographical features. This makes the paper a bit difficult to read.**

In the revised manuscript, better clarifications are given in order to avoid any confusion to the reader. In our study we have used four datasets (MODIS-Aqua, CALIOP, MERRA-2 and AERONET) which are utilized as follows. MODIS-Aqua AOD and MDF (i.e., MERRA-2) are combined in order to obtain MIDAS DOD. The evaluation of MDF is made against LIVAS, which has been developed based on the CALIOP profiles. MIDAS

DOD is compared versus MERRA-2 and LIVAS DODs while it has been evaluated against AERONET DODs, extracted according to the methodology described in Section 2.4.

**7. Many numbers are given with too many digits. Please try to only provide significant digits.**

We have reduced the number of decimal digits.

**Major specific comments:**

1. Line 33: "ground-truth AERONET-derived DODs" -> There is no "truth", any measurement has uncertainties and biases. In particular, here the DOD derived from AERONET is a complex product with a number of assumptions and no-one should see it as "the truth".

We agree that the word "truth" is not appropriate and we have remove it.

2. Section 2.3 on CALIOP: It is a bit unclear to me how the CALIOP subtypes are used in LIVAS. I have the feeling that LIVAS is a different retrieval, not using the CALIOP "official" features/retrievals and therefore I would recommend to only mention here what is really needed to understand LIVAS. Otherwise it is a bit confusing.

Section 2.3 consists of two paragraphs where in the first one, a brief discussion about CALIOP retrievals and products is given while in the second one the main steps for the derivation of the post-processed CALIOP-related pure dust product, available from the ESA-LIVAS database, maintained and hosted at the National Observatory of Athens (NOA), are intentionally briefly described because all the relevant publications (Amiridis et al., 2013; 2015; Marinou et al., 2017; Proestakis et al., 2018) are already provided. A short comment that may can help in the clarification of any misleading points. LIVAS is the outcome of the post-processing of CALIOP profiles by applying: (i) "corrections" on dust lidar ratio (Amiridis et al., 2013), (ii) a discrimination technique for "extracting" dust aerosols from dusty mixtures, (iii) a series of quality control filters (Marinou et al., 2017) and (iv) aggregation into Level 3 outputs (Tackett et al., 2018).

**3. Section 3.1 on the methodology: Do you see a discontinuity in the MODIS DOD linked to changing of MERRA-2 grid cell?**

The spatial variations found in MIDAS DOD are attributed to those of MODIS AOD since the MERRA-2 grid cell has not been changed for the derivation of the fine resolution dataset. For the intercomparison among MIDAS, LIVAS and MERRA-2, all datasets have been regridded to 1° x 1° pixels for consistency reasons without noticing pronounced features, discontinuities and abrupt changes in the DOD geographical patterns.

4. Lines 330 to 332: "Our approach avoids on purpose the inclusion of additional optical properties providing information on aerosol size (alpha) available from MODIS and absorptivity (Aerosol Index) from OMI that are characterized by inherent limitations". -> This is probably very unclear for the non-specialist reader and comes a bit out of the blue. If OMI is mentioned, the authors should at least explicit why using OMI would make sense when looking for dust aerosols and what are the potential drawbacks with it, other than data availability.

We have added a reference (Torres et al., 1998) describing the theoretical background. We think that this clarifies the published approach that positive AI values are associated with the presence of absorbing mineral particles. Even higher AI levels are found when biomass particles are probed. There are numerous studies which have been relied on AI thresholds (usually above 1) for the monitoring and tracking loads of absorbing particles. The theoretical background of the UV retrievals is given in Torres et

al. (1998) while updates of the applied OMI algorithm are described in Torres et al. (2013). Likewise, a nice overview of the OMI products can been found in Torres et al. (2007).

5. Lines 359-361: "Here, we are using the same equations replacing AERONET AODs with those given by MODIS. This relies on the fact (results not shown here) that their averages are almost unbiased." -> A bit unclear. Average of MODIS AODs unbiased wrt AERONET? If yes, this feels a bit short and at least a reference to the MODIS validation should be given and this should be discussed. A quick search (https://doi.org/10.1016/j.atmosenv.2018.12.004) showed me that indeed on average along many years and globally the MODIS mean bias wrt AERONET is particularly low. However, at regional scale this is not always true. In particular, over the dustiest regions (N Africa and Middle East) it seems that there are more outliers in the comparisons linked to a more difficult AOD retrieval from MODIS. I am not saying that the MODIS AOD should not be used to estimate the uncertainty on the AOD, but that it should be discussed a bit more.

We have rephrased this sentence as follows.

"This relies on the fact (results not shown here) that their averages from a global perspective are almost unbiased; however, at regional level, small negative or positive offsets (lower than 0.05 in absolute terms) are recorded in the vast majority of AERONET sites, thus supporting our argument."

6. Equations 4 and 5: Those do confirm my thoughts in the previous comment. Levy et al (2013) write that (for DT land) "69.4% of MODIS AOD fall within expected uncertainty of ±(0.05 + 15 %)." This is, I guess, the origin of equation 5 here (and something similar can be found for equation 4). This means, to me, that DT land has a mean bias of 0.05 wrt AERONET, in contradiction with the sentence above saying that there is no bias.

Please see our previous reply.

Equation 8: I would appreciate a plot here (of the data that lead to this equation), both to show how good the fit is (does it really need such a complex polynomial curve?) and show some values. I computed them myself from the equation to have a feeling. [MDF uncertainty]: [0 0,2]; [0,1 0,14]; [0,2 0,15]; [0,3 0,18]; [0,4 0,22]; [0,5 0,25]; [0,6 0,26]; [0,7 0,24]; [0,8 0,18]; [0,9 0,1]; [1 -0,01]; Those uncertainties are not negligible (especially at low MDF), however they are not discussed at all.

The requested plot along with the relevant text (copied from the submitted manuscript) are provided below.

"The CALIOP DOD-to-AOD ratio is our reference for estimating the uncertainty limits of the MERRA-2 dust fraction (MDF). The analysis is performed at 1° x 1° spatial resolution considering only grid cells in which both MERRA-2 and CALIOP DODs are higher to or equal than 0.02. According to this criterion, more than 450000 CALIOP-MERRA2 collocated pairs have been found which are sorted (ascending order) based on MERRA-2 MDF (ranging from 0 to 1) and then are grouped in equal size bins containing 20000 data each sub-sample. For every group, we computed the median MDF (x axis) as well as the 68th percentile of the absolute MERRA-2 – CALIOP dust fraction (y axis) and then we found the best polynomial fit (Eq. 8)."

The geographical distributions illustrated in Figs. 1 and 2, along with the plot given here, indicate that the MDF uncertainties are not negligible and at the same time reveal that they are higher over areas where the dust loads do not dominate in the total burden. Based on the uncertainty analysis, it is apparent that the relative error gradually decreases from ~50% to ~10%, for increasing dust fraction, when MDF values higher than 0.5 (or 50%) are considered. For lower MDF levels (

8. Section 4.1 (and also in the conclusion): I do not at all understand this section. Why should the MODIS and MERRA-2 AODs be different? That would underline problems in one of the data sets (at least). And why do they need to be different for this analysis to work? I would say the opposite, that MODIS and MERRA-2 AODs should be similar enough to allow for this work to be relevant. Overall, I find this section 4.1 quite confusing, I am unsure what the authors are trying to show and how it fits in the rest of the paper. I would better see here a short summary of the MERRA-2 AOD validation (with references). And then a short discussion how this will impact the MIDAS data set. Also, this section contains discussions linked to the MDF (dust emission in GOCART for example), which should be moved to the next section.

As suggested by the Reviewer, we have totally removed Section 4.1 from the revised manuscript and we have kept only parts which fit to the revised text.

**9. Lines 455 to 457: Is the MIDAS DOD expected to be overestimated because the GOCART model overestimates dust emissions?**

In theory yes but that is why we decided to use also LIVAS. In addition, it is not easy to give a sufficient answer because the number of AERONET stations in dust emission areas are limited. For example, in N. Africa, we found slight positive bias [0.0, 0.02] in the Tamanrasset site. Regarding dust transport, there are several factors (e.g., size distribution, winds, identification and activation of the sources, etc.) that can affect the representation of dust burdens by MERRA-2. According to the global DOD maps (Figs. 5 and S6 in the revised manuscript), the agreement of MERRA-2 with respect to LIVAS and MIDAS is good both on annual and seasonal scales. We would like to point out again that from MERRA-2 we are using the dust fraction which is determined by the ability of the model to reproduce accurately both dust and non-dust aerosol species.

**10. Figure 3: High resolution figures are needed. Here if I zoom in (to see details discussed) it becomes blurry**

The quality of the produced png files might have been affected by the conversion of the word file to pdf. However, in order to improve the illustration, we have increased the dpi from 300 to 600.

**11. Lines 485 to 490: I don't understand. If there is a bias of about 10% but other metrics show the algorithm performs well, then why is there a bias? This should be explained also in the manuscript.**

By default, mean bias is affected by outliers in contrast to FB and FGE which are more "smoothed". Some short clarifications along with a reference of the evaluation metrics have been added in the revised manuscript.

12. Lines 493-494: the correlation between MERRA-2 and CALIOP (LIVAS??) is less good over dust source regions due to the high variability. This is linked by the authors to a poor behaviour of the model in these cases. Can't we also imagine that CALIOP is not perfect there, as the very thin ground coverage makes it miss many events? This is discussed a bit further (lines 521 onwards), but I think it would be good to also mention around lines 493-494 that CALIOP (LIVAS?) is also not perfect.

We don't think that the narrow footprint of CALIOP at the ground plays such an important role since dust plumes are in general spatially wide (i.e., covering a whole 1-degree grid cell). Definitely CALIOP does not provide the "perfect retrievals" and the main reasons causing a departure from the "ideal scenario" are mentioned mainly in Section 4.2 as well as in relevant parts of the text.

13. Lines 533-534: the underestimation of CALIOP with respect to AERONET, is it the official product or LIVAS? Here, in this section, it is very confusing. I think most of the section refers to LIVAS but this specific sentence to the official product. If this is indeed the case, then I do not see how this information (and the discussion following) is useful here in the paper, where LIVAS and MERRA-2 are compared. That discussion is already in section 2.3 in a different formulation.

Between lines 533 and 542 are mentioned two important factors which can hamper CALIOP's performance in reproducing columnar DOD. The first one is related to the dust lidar ratio (LR), which has to be used for the derivation of the extinction profiles (resulted from the multiplication of backscatter coefficients with lidar ratio) and subsequently are vertically integrated in order to get the columnar DOD. This is what actually has been proposed in Amiridis et al. (2013) while in the global map of Figure S1 are depicted the most "representative" lidar ratios applied in our study. The second factor is related to the total attenuation of the laser beam by mineral particles accumulated at very high concentrations and this means that the columnar DOD will be underestimated under these cases since there are not available retrievals throughout and beneath the very thick dust layer (Konsta et al., 2018). Summarizing, the impact of the first factor has been mitigated in the LIVAS database (i.e., selection of more realistic LR than the universal values used in the raw CALIOP data) while for the second one cannot be done nothing.

**14. Lines 558-560: Does this mean that overall, only 10 to 20 CALIOP measurements per grid cell were averaged along 9 years? If yes, this is very low and I don't think it can be considered representative.**

The number of CALIOP L2 profiles (5km resolution) aggregated for the derivation of each LIVAS 1° x 1° grid cell varies from 1 to 24. Overall, we have ~3.4 million of MERRA2-LIVAS pairs (i.e., total number of collocated 1° x 1° grid cells) and we show how these are distributed among classes defined based on the number of CALIOP L2 profiles falling within the 1° x 1° grid cell. A short clarification has been added in the revised manuscript and it is given below.

*"Figure S3-ii displays the long-term averaged geographical distribution of the number of CALIOP L2 profiles (up to 24) aggregated for the derivation of the LIVAS 1°x1° grid-cell."*

15. General on section 4.2: this section is quite long, it contains the description of the differences and some discussion about the origin of those differences, but no discussion on the implications of underlined shortcomings on the MIDAS data set? In particular, the underestimation of MERRA-2 over dust sources should be discussed in terms of "how will it affect the MIDAS DOD".

As we wrote in our introductory response, we have made an effort to reduce the manuscript, shortening also Section 4.2. Regarding MERRA-2 underestimations over dust sources, we think that our reply in a similar comment mentioned above is valid also here. Focusing only over sources, the underestimation of dust emission by MERRA-2 most probably has minor impact on our results since the amount of mineral particles will be lower but their portion (i.e., MDF) to the total load (in optical terms) will be rather stable and above ~90% considering that the contribution from other aerosol species is very small or even zero. As it concerns dust transport, the situation becomes more complex because an accurate representation of the MDF is determined both from dust and non-dust AODs. Actually, our findings (Section 4.1) can be the starting point of a dedicated assessment analysis in which all the factors that affect MERRA-2 dust fraction will be investigated in-depth. Finally, we would like to express our disagreement with the comment "..., but no discussion on the implications of underlined shortcomings on the MIDAS data set?". There are several "links" in Sections 4.2 and 4.3 with the obtained findings from Section 4.1 facilitating the interpretation of the deviations found between MIDAS and AERONET (Section 4.2) as well as the disagreements among MIDAS, LIVAS and MERRA-2 DODs (Section 4.3).

**16. Section 4.3: Why redo a MODIS validation against AERONET (not bringing anything new)? I think there are enough papers on that to just refer to one and remove this part, making the paper a bit shorter and less confusing.**

We have removed the relevant part in the revised manuscript.

17. Figure 5: please change the colour scale for the correlation coefficient as it is now very difficult to see

Done.

18. Section 4.4: I think that somehow this section should show what the new MIDAS product brings. The comparisons are currently done in a way that gives the impression it's just another product but not really improved or different from MERRA-2 or the LIVAS climatology. This is linked to the fact that averages over long periods are analysed, so at the end we are just comparing (validating?) climatologies from different products. As MIDAS is not meant to be a climatology, I would not do this kind of comparisons a big point in the paper, but I would instead emphasize what MIDAS gives that those other products can't give. And validate the product at its resolution - but this is done in the comparison with AERONET.

The powerful elements of the MIDAS dataset, as already have been mentioned in the submitted text, are the almost global coverage, the long-term availability as well as the fine spatial resolution. These features are demonstrated in Section 4.4 where a short discussion about the annual and seasonal global DOD patterns is given. A more detailed climatological analysis is under preparation whereas other studies, relying on the MIDAS dataset, are ongoing and they have already mentioned in the abstract (Lines 35-37), in the introduction (Lines 129-132) and in the last paragraph of the summary.

In Section 4.3, our intention is to make an intercomparison among MERRA-2, LIVAS and MIDAS DODs aiming at assessing the consistency of our product with respect to those provided by simulations and active remote sensing techniques. To our opinion this a very important aspect in our study taken into account the advantages/disadvantages of each dataset. Based on this analysis, we highlighted differences found at specific regions of the planet while through the intra-annual plots for each subdomain the obtained deviations among the three datasets are illustrated and interpreted. Finally, the length of the section has been reduced after refining the submitted document.

**19. Line 648-649: "the study period extends from 2007 to 2015, driven again from CALIOP's temporal availability" -> this is very confusing. . . CALIOP is still running. . . so the authors probably mean**

**LIVAS availability. This is only one of the many examples where it is not clear which data is referred to, leaving the reader in possible misunderstanding.**

We agree with the Reviewer and the appropriate corrections have been made.

**20. Figure 6: Why do we see orbit-like features on a 9 years average?**

Please note that CALIOP revisiting time is 16 days and it has a narrow footprint at the ground. In addition, a series of quality checks is applied on the raw vertical profiles. Therefore, it makes sense that orbit-like features are evident on the long-term averaged global distributions even though these are representative for a 9-year period. In the following figure, the patterns of the dust fraction at daytime (first) and nighttime (second) conditions as well as throughout the day (third), are given. It is clarified that the same CALIOP data have been used for the reproduction of Figure 6 (in the submitted manuscript) but here a different parameter is processed. It is clear that the orbit-like features are evident when only daytime or nighttime patterns are studied in contrast to the "full-day" distribution.